# Let There be Direction in Hypergraph Neural Networks

**Stefano Fiorini**                                                                                    *stefano.fiorini@iit.it*
*Pattern Analysis & Computer Vision (PAVIS)*
*Italian Institute of Technology (IIT)*

**Stefano Coniglio**                                                                                  *stefano.coniglio@unibg.it*
*Department of Economics*
*University of Bergamo*

**Michele Ciavotta**                                                                                  *michele.ciavotta@unimib.it*
*Department of Informatics, Systems and Communication*
*University of Milano-Bicocca*

**Alessio Del Bue**                                                                                   *alessio.delbue@iit.it*
*Pattern Analysis & Computer Vision (PAVIS)*
*Italian Institute of Technology (IIT)*

**Reviewed on OpenReview:** *https://openreview.net/forum?id=h48Ri6pmvi*

## Abstract

Hypergraphs are a powerful abstraction for modeling high-order interactions between a set of entities of interest and have been attracting a growing interest in the graph-learning literature. In particular, directed hypegraphs are crucial in their capability of representing real-world phenomena involving group relations where two sets of elements affect one another in an asymmetric way. Despite such a vast potential, an established, principled solution to tackle graph-learning tasks on directed hypergraphs is still lacking. For this reason, in this paper we introduce the *Generalized Directed Hypergraph Neural Network* (GeDi-HNN), the first spectral-based Hypergraph Neural Network (HNN) capable of seamlessly handling hypergraphs featuring both directed and undirected hyperedges. GeDi-HNN relies on a graph-convolution operator which is built on top of a novel complex-valued Hermitian matrix which we introduce in this paper: the *Generalized Directed Laplacian* $\vec{L}_N$. We prove that $\vec{L}_N$ generalizes many previously-proposed Laplacian matrices to directed hypergraphs while enjoying several desirable spectral properties. Extensive computational experiments against state-of-the-art methods on real-world and synthetically-generated datasets demonstrate the efficacy of our proposed HNN. Thanks to effectively leveraging the directional information contained in these datasets, GeDi-HNN achieves a relative-percentage-difference improvement of 7% on average (with a maximum improvement of 23.19%) on the real-world datasets and of 65.3% on average on the synthetic ones.

## 1 Introduction

In recent years, ground-breaking research in the graph-learning literature has been prompted by seminal works on Graph Neural Networks (GNNs) such as (Scarselli et al., 2009; Micheli, 2009; Li et al., 2016; Kipf and Welling, 2017; Veličković et al., 2018). Since representing a set of complex relationships solely through undirected or directed graphs can prove too restrictive in many real-world scenarios, generalizations to graphs allowing for higher-order (group) relationships, i.e., hypergraphs, have been considered. Hypergraphs generalize the notion of a graph to the case where an edge (a hyperedge) can connect an arbitrary number of nodes, thus allowing to capture not just pairwise (dyadic) relationships but also group-wise (polyadic) dynamics (Schaub et al., 2021). This has led to a new stream of research devoted to the investigation of

Hypergraph Neural Networks (HNNs) (Feng et al., 2019; Chien et al., 2021; Huang and Yang, 2021; Wang et al., 2023a;b).

While most of the literature on HNNs has focused on undirected hypergraphs, many real-world phenomena such as chemical reactions are naturally modeled on hypergraphs whose hyperedges have a notion of direction. Despite this, the directed case has been addressed only sporadically, and often only in application-specific scenarios such as traffic forecasting (Luo et al., 2022) and music recommendation (La Gatta et al., 2022). To the best of our knowledge, a general solution based on a convolution operator which is solidly grounded in spectral graph theory while not being problem-dependent is missing. We aim at bridging such a gap.

In this paper, we introduce the *Generalized-Directed Hypergraph Neural Network* (GeDi-HNN), the first spectral-based HNN capable of seamlessly handling hypergraphs featuring both directed and undirected hyperedges. GeDi-HNN relies on a graph-convolution operator which is built on top of a novel Hermitian Laplacian matrix which we introduce in this paper: the *Generalized Directed Laplacian* $\vec{L}_N$. $\vec{L}_N$ generalizes various Laplacian matrices: the one proposed in Zhou et al. (2006) and used in Feng et al. (2019) for undirected hypergraphs, the Sign-Magnetic Laplacian proposed for directed graphs in (Fiorini et al., 2023), and the classical Laplacian matrix (Chung and Graham, 1997) used for undirected graphs.

**Main Contributions of the Work**

- We extend the literature on spectral-based HNNs by introducing GeDi-HNN, the first spectral HNN capable of handling hypergraphs with both directed and undirected hyperedges.

- We introduce the *Generalized Directed Laplacian* matrix $\vec{L}_N$; we prove that it enjoys several desirable properties, among which admitting an eigenvalue decomposition, and that it generalizes many existing Laplacian matrices.

- Compared to state-of-the-art methods, GeDi-HNN achieves a relative-percentage-difference improvement of 7% on average (with a maximum improvement of 23.19%) on the real-world datasets and of 65.3% on average on the synthetic ones. This demonstrates its efficacy in extracting and utilizing the information encoded in the hyperedge directions.

The paper is organized as follows. Preliminaries and previous works are summarized in Section 2. $\vec{L}_N$ is introduced in Section 3, along with its properties. Section 4 provides an overview of GeDi-HNN's architecture, which is built upon $\vec{L}_N$. Experimental results are reported in Section 5. Conclusions are drawn in Section 6. The proofs of our theorems and additional details are provided in the Appendix.

## 2 Preliminaries and Previous Work

### Undirected and Directed Hypergraphs

A hypergraph is an ordered pair $\mathcal{H} = (V, E)$, with $n := |V|$ and $m := |E|$, where $V$ is the set of vertices (or nodes) and $E \subseteq 2^V \setminus \{\}$ is the (nonempty) set of hyperedges. The hyperedge weights are stored in the diagonal matrix $W \in \mathbb{R}^{m \times m}$. The vertex and hyperedge degrees are defined as $d_u = \sum_{e \in E : u \in e} |w_e|$ for $u \in V$ and $\delta_e = |e|$ for $e \in E$ and are stored in two diagonal matrices $D_v \in \mathbb{R}^{n \times n}$ and $D_e \in \mathbb{R}^{m \times m}$. Hypergraphs where $\delta(e) = k$ for some $k \in \mathbb{N}$ for all $e \in E$ are called $k$-uniform. Graphs are 2-uniform hypergraphs. Following Gallo et al. (1993), we define a directed hypergraph as a hypergraph where each edge $e \in E$ is partitioned in a *head set* $H(e)$ and a *tail set* $T(e)$. If $T(e)$ is empty, $e$ is an undirected edge.

### Graph Fourier Transform and Graph Convolutions

Let $\mathcal{L}$ be a generic Laplacian matrix of a given 2-uniform hypergraph $\mathcal{H}$ which embeds its topology. We assume that $\mathcal{L}$ admits an eigenvalue decomposition $\mathcal{L} = U \Lambda U^*$, with $U \in \mathbb{C}^{n \times n}$. $U^*$ is the conjugate transpose of $U$, and $\Lambda \in \mathbb{R}^{n \times n}$ is a diagonal matrix. Let $x \in \mathbb{C}^n$ be a *graph signal*, i.e., a function $x : V \to \mathbb{C}$ whose domain coincides with the vertices of $\mathcal{H}$. Following Shuman et al. (2013), we call $\hat{x} = \mathcal{F}(x) = U^* x$ the *graph*

*Fourier transform* of $x$ and $\mathcal{F}^{-1}(\hat{x}) = U\hat{x}$ its inverse transform. The convolution $y \circledast x$ between $x$ and another graph signal $y \in \mathbb{C}^n$ (taking the role of a *filter*) has a natural construction in the frequency space, where it is defined as $y \circledast x = U \text{diag}(U^*y)U^*x$. Letting $\hat{Y} := U\hat{G}U^*$ with $\hat{G} := \text{diag}(U^*y)$, we can write $y \circledast x$ in the vertex space as the linear operator $\hat{Y}x$.

In the context of a GNN, there are two drawbacks to learning $y$ explicitly as a *non-parametric filter*: *i)* deriving the eigenvalue decomposition of $\mathcal{L}$ could be computationally too intensive (Kipf and Welling, 2017); *ii)* learning $y$ explicitly would require learning a number of parameters proportional to the input size, which could be inefficient for tasks of high dimension (Defferrard et al., 2016).

For these reasons, it is customary in the GNN literature, see Kipf and Welling (2017) and Defferrard et al. (2016), to work with filters whose graph Fourier transform is a degree-$K$ polynomial function of $\Lambda$ with a small $K$. This leads to a so-called *localized filter* thanks to which the output (i.e., filtered) signal at a vertex $u \in V$ is a linear combination of the input signals within a $K$-hop neighborhood of $u$ (Shuman et al., 2013). Using either Chebyshev polynomials as done by Hammond et al. (2011) and Kipf and Welling (2017) or power monomials as done by Singh and Chen (2022), with $K = 1$ (as typical in the literature) one obtains a parametric family of linear operators with two (learnable) parameters $\theta_0$ and $\theta_1$:[1]

$$\hat{Y} := \theta_0 I + \theta_1 \mathcal{L}. \tag{1}$$

**Discrete Laplacians for Undirected Hypergraphs**

In a hypergraph $\mathcal{H} = (V, E)$, the relationship between vertices and hyperedges is classically represented via an incidence matrix $B$ of size $|V| \times |E|$. When $\mathcal{H}$ is undirected, $B$ is defined as:

$$B_{ve} = \begin{cases} 1 & \text{if } v \in e \\ 0 & \text{otherwise} \end{cases} \qquad v \in V, e \in E. \tag{2}$$

From $B$, one can derive $Q$, the *Signless Laplacian Matrix* Chung and Graham (1997), as well as its normalized counterpart $Q_N$:

$$Q := BWB^\top \qquad\qquad Q_N := D_v^{-\frac{1}{2}} BW D_e^{-1} B^\top D_v^{-\frac{1}{2}}. \tag{3}$$

When restricting to undirected graphs (i.e., 2-uniform undirected hypergraphs), an alternative Laplacian matrix, the so-called *Signed Laplacian Matrix*, can be obtained with a similar construction to equation 3. This involves applying an arbitrary orientation to the edges of the graph (i.e., arbitrarily multiplying by $-1$ exactly one entry per column of $B$). Calling such a matrix $B'$, the *Signed Laplacian matrix* $L$ and its normalized counterpart $L_N$ are defined as follows:

$$L := B'WB'^\top \qquad\qquad L_N := D_v^{-\frac{1}{2}} B'W D_e^{-1} B'^\top D_v^{-\frac{1}{2}}. \tag{4}$$

By utilizing the standard definitions of weighted adjacency matrix $A \in \mathbb{R}^{n \times n}$ where $A_{uv} = w_e$ if $e = \{u, v\} \in E$ and $A_{uv} = 0$ otherwise, for undirected graphs we have:

$$Q = D_v + A \qquad L = D_v - A \qquad Q_N = I - L_N \qquad L_N = I - Q_N. \tag{5}$$

While the definition of $L$ in equation 4 does not extend nicely to general (not 2-uniform) hypergraphs, the definition of $L_N$ in equation 5 does.[2] A generalization of the *Signed Laplacian* to general undirected hypergraphs which follows $L_N = I - Q_N$ from equation 5 is proposed by Zhou et al. (2006), and reads:[3]

$$\Delta = I - Q_N. \tag{6}$$

---

[1]Following w.l.o.g. Singh and Chen (2022), we employ the approximation $\hat{G} = \sum_{k=0}^K \theta_k \Lambda^k$, from which we deduce $\hat{Y}x = U\hat{G}U^*x = U(\sum_{k=0}^K \theta_k \Lambda^k)U^*x = \sum_{k=0}^K \theta_k(U\Lambda^k U^*)x = \sum_{k=0}^K \theta_k \mathcal{L}^k x$.

[2]For instance, the choice of which entries of $B$ should be multiplied by $-1$ would drastically affect $L$ (rendering the orientation not arbitrary anymore) and $L$ may feature both positive and negative off-diagonal entries, thereby violating $L_N = I - Q_N$ (notice that $Q_N \geq 0$ holds by construction).

[3]In Zhou et al. (2006), $Q_N$ is called $\Theta$.

Notably, all the Laplacian matrices we introduced satisfy some key properties: *i)* they are real and symmetric—and thus diagonalizable with real-valued eigenvalues; *ii)* they are positive semi-definite; and *iii)* their normalized versions possess a bounded spectrum.

**Discrete Laplacian Matrices for Directed 2-Uniform Hypergraphs**

In directed 2-uniform hypergraphs, the presence of edge directions renders the graph asymmetric and none of the previous definitions of the graph Laplacian apply. Indeed, those in equation 3, equation 4, and equation 6 would symmetrize the graph and destroy its directions, while the one in equation 5 would lead to an asymmetric matrix which does not admit an eigenvalue decomposition and, thus, would prevent the application of the graph Fourier transform.

The *Magnetic Laplacian* $L^{(q)}$, proposed by Lieb and Loss (1993) in the context of electromagnetic fields and adopted within a spectral GNN by Zhang et al. (2021b;a), is a complex-valued, Hermitian Laplacian matrix. It encodes the directional information of the graph while enjoying an eigenvalue decomposition with a nonnegative, real spectrum. This Laplacian matrix generalizes the Laplacian $L$ defined in equation 5. Letting $A_s := \frac{1}{2}\left(A + A^\top\right)$ be the symmetrized version of $A$ and letting $D_s := \mathrm{diag}(A_s\,\boldsymbol{e})$, where $\boldsymbol{e}$ is the all-one vector, the *Magnetic Laplacian* and its normalized version are defined as follows:

$$L^{(q)} := D_s - H^{(q)} \text{ and } L_N^{(q)} := I - D_s^{-\frac{1}{2}} H^{(q)} D_s^{-\frac{1}{2}}, \text{with } H^{(q)} := A_s \odot \exp\left(\mathrm{i}\,2\pi q\left(A - A^\top\right)\right),$$

where i is the imaginary unit and $q \in [0,1]$.

The *Sign-Magnetic Laplacian* $L^\sigma$ is a matrix proposed by Fiorini et al. (2023) which is well-defined also for graphs with negative edge weights and enjoys some extra desirable properties. If $q = \frac{1}{4}$, $L^\sigma$ and $L^{(q)}$ coincide if the latter is first constructed for an unweighted version of the graph and then multiplied component-wise by $A_s$. Thus, $L^\sigma$ is invariant to a positive weight scaling which could otherwise alter the sign pattern of $L^{(q)}$ and, thus, the edge direction. Letting $\bar{D}_s := \mathrm{diag}(|A_s|\,\boldsymbol{e})$ and $\mathrm{sign} : \mathbb{R} \to \{-1,0,1\}$ be the component-wise *signum* function, $L^\sigma$ and its normalized version are defined as follows:

$$L^\sigma := \bar{D}_s - H^\sigma \text{ and } L_N^\sigma := I - \bar{D}_s^{-\frac{1}{2}} H^\sigma \bar{D}_s^{-\frac{1}{2}}, \text{ with } H^\sigma := A_s \odot \left(e^\top - \mathrm{sign}(|A - A^\top|) + \mathrm{i}\,\mathrm{sign}\left(|A| - |A^\top|\right)\right).$$

To the best of our knowledge, no extensions of the Laplacian matrix are known for the case of directed hypergraphs which are not 2-uniform. Our paper aims to bridge this gap.

## 3 The Generalized Directed Laplacian

We now introduce our proposed complex-valued Hermitian Laplacian matrix, which is capable of handling hypergraphs featuring both directed and undirected hyperedges. We also establish some of its key properties. We refer to this matrix as the *Generalized Directed Laplacian*. We define it directly in normalized form as:

$$\vec{L}_N := I - \vec{Q}_N \qquad \text{with} \qquad \vec{Q}_N := D_v^{-\frac{1}{2}} \vec{B} W D_e^{-1} \vec{B}^* D_v^{-\frac{1}{2}}, \tag{7}$$

where $\vec{B}$ is the following complex-valued incidence matrix:

$$\vec{B}_{ve} = \begin{cases} 1 & \text{if } v \in H(e) \\ -\mathrm{i} & \text{if } v \in T(e) \qquad v \in V, e \in E. \\ 0 & \text{otherwise} \end{cases} \tag{8}$$

To appreciate how $\vec{L}_N$ encodes the directions of the hypergraph, we analyze its scalar form for a pair of vertices $u, v \in V$:

$$(\vec{L}_N)_{uv} = \begin{cases} 1 - \displaystyle\sum_{e \in E : u \in e} \frac{w_e}{\delta_e} \frac{1}{d_u} & u = v \\[4ex] \left(-\displaystyle\sum_{\substack{e \in E: \\ u,v \in H(e) \vee u,v \in T(e)}} \frac{w_e}{\delta_e} - \mathrm{i}\left(\displaystyle\sum_{\substack{e \in E: \\ u \in T(e) \wedge v \in H(e)}} \frac{w_e}{\delta_e} - \displaystyle\sum_{\substack{e \in E: \\ u \in H(e) \wedge v \in T(e)}} \frac{w_e}{\delta_e}\right)\right) \dfrac{1}{\sqrt{d_u}} \dfrac{1}{\sqrt{d_v}} & u \neq v \end{cases} \tag{9}$$

The pair $u, v$ affects the value of $(\vec{L}_N)_{uv}$ through each hyperedge $e \in E$ where $u, v \in e$. Considering the second line of equation 9, each hyperedge where $u, v$ take both the role of head ($u, v \in H(e)$) or tail ($u, v \in T(e)$) contributes negatively to the real part, $\Re((\vec{L}_N)_{uv})$, by the opposite of its normalized weight ($-w_e/\delta_e$). For undirected hypergraphs, this is the only contribution. Such a behavior is in line with the nature of $L_N$ (equation 4) for undirected graphs and of $\Delta$ (equation 6) for undirected hypergraphs. Hyperedges where $u, v$ take opposite roles contribute with their normalized weight negatively if $u \in H(e)$ and $v \in T(e)$ and positively if $u \in T(e)$ and $v \in H(e)$. Due to this, the imaginary part, $\Im((\vec{L}_N)_{uv})$, is affected by the *net* contribution of $u$ and $v$ across all the directed hyperedges that contain them. This is in line with the *net flow* behavior observed by (Fiorini et al., 2023) for $L^\sigma$ for the case of directed graphs. An example illustrating the construction of $\vec{L}_N$ for a directed hypergraph is provided in Appendix F.

The relationship between $\vec{L}_N$ and the previously-proposed Laplacian matrices that we introduced in Section 2 as well as its spectral properties are analyzed in more detail in what follows.

## On the Relationship between $\vec{L}_N$ and other Laplacian Matrices

The following theorem shows that $\vec{L}_N$ and $\vec{Q}_N$ generalize the Laplacian matrices proposed by Zhou et al. (2006) for undirected hypergraphs which are defined in equation 3 and equation 6:

**Theorem 1.** *If $\mathcal{H}$ is an undirected hypergraph, $\vec{L}_N = \Delta$ and $\vec{Q}_N = Q_N$.*

Focusing on 2-uniform hypergraphs, we show that $\vec{L}_N$ and $\vec{Q}_N$ generalize the Signed and Signless Laplacian matrices $L_N$ and $Q_N$ defined in equation 3 and equation 4, which are classically used for undirected graphs (Chung and Graham, 1997):

**Corollary 1.** *If $\mathcal{H}$ is an undirected 2-uniform hypergraph, $\vec{L}_N = \frac{1}{2}L_N$ and $\vec{Q}_N = \frac{1}{2}Q_N$.*

Focusing on the case of directed graphs, we establish under which conditions $\vec{L}_N$ generalizes the Signum-Magnetic Laplacian proposed by Fiorini et al. (2023) and the Magnetic Laplacian introduced by Lieb and Loss (1993):

**Theorem 2.** *If $\mathcal{H}$ is a directed 2-uniform hypergraph with no antiparallel edges, we have $\vec{L}_N = \frac{1}{2}L_N^\sigma$ with $A_s = A + A^\top$.*

**Corollary 2.** *If $\mathcal{H}$ is a directed 2-uniform unweighted hypergraph with no antiparallel edges, we have $\vec{L}_N = \frac{1}{2}L_N^{(q)}$ with $q = \frac{1}{4}$ and $A_s = A + A^\top$.*

## Key Spectral Properties of $\vec{L}_N$

We start by showing that $\vec{L}_N$ and $\vec{Q}_N$ admit an eigenvalue decomposition with real eigenvalues. The result is structural and follows after showing that both matrices are Hermitian:

**Theorem 3.** *$\vec{L}_N$ and $\vec{Q}_N$ are diagonalizable with real eigenvalues.*

Next, we show that the spectrum of $\vec{Q}_N$ is nonnegative. This result is obtained by showing that $\vec{Q}_N$ can be decomposed into the product of the matrix $D_v^{-\frac{1}{2}}\vec{B}W^{\frac{1}{2}}D_e^{-\frac{1}{2}}$ and its conjugate transpose:

**Theorem 4.** *$\vec{Q}_N$ is positive semidefinite.*

To show that $\vec{L}_N$ is positive semidefinite, we first derive the equation of $||x||^2_{\vec{L}_N}$, i.e., the $p$-Dirichlet energy function with $p = 2$ induced by the Generalized Directed Laplacian for a signal $x \in \mathbb{C}^n$. In line with the 2-uniform case (Shuman et al., 2013), such a function provides a measure of the global smoothness of $x$ across the entire hypergraph.

**Theorem 5.** *Let $x = a + ib \in \mathbb{C}^n$, with $a, b \in \mathbb{R}^n$. The 2-Dirichlet energy function $||x||_{\vec{L}_N}^2 = x^* \vec{L}_N x$ of $x$ induced by $\vec{L}_N$ is the following quadratic form:*

$$\frac{1}{2} \sum_{e \in E} \frac{w(e)}{\delta(e)} \sum_{u,v \in E} \left( \left( \left( \frac{a_u}{\sqrt{d_u}} - \frac{a_v}{\sqrt{d_v}} \right)^2 + \left( \frac{b_u}{\sqrt{d_u}} - \frac{b_v}{\sqrt{d_v}} \right)^2 \right) \mathbf{1}_{u,v \in H(e) \vee u,v \in T(e)} \right.$$
$$+ \left( \left( \frac{a_u}{\sqrt{d_u}} + \frac{b_v}{\sqrt{d_v}} \right)^2 + \left( \frac{a_v}{\sqrt{d_v}} - \frac{b_u}{\sqrt{d_u}} \right)^2 \right) \mathbf{1}_{u \in H(e), v \in T(e)}$$
$$\left. + \left( \left( \frac{a_u}{\sqrt{d_u}} - \frac{b_v}{\sqrt{d_v}} \right)^2 + \left( \frac{a_v}{\sqrt{d_v}} + \frac{b_u}{\sqrt{d_u}} \right)^2 \right) \mathbf{1}_{v \in H(e), u \in T(e)} \right), \tag{10}$$

*where $\mathbf{1}$ is the indicator function.*

**Corollary 3.** *$\vec{L}_N$ is positive semidefinite.*

Following the methods used in (Kipf and Welling, 2017; Zhang et al., 2021b; Fiorini et al., 2023), we show that our Laplacian is positive semidefinite and thus can be adopted as a convolutional operator.

Lastly, we combine Corollary 3 and Theorem 4 to derive upper bounds on the spectra of $\vec{L}_N$ and $\vec{Q}_N$:

**Corollary 4.** *$\lambda_{\max}(\vec{L}_N) \leq 1$ and $\lambda_{\max}(\vec{Q}_N) \leq 1$.*

While these spectral bounds are not required for the construction of the convolution operator defined in equation 1, they are necessary to construct localized filters using Chebyshev's polynomials of degree $K > 1$ (Kipf and Welling, 2017; Defferrard et al., 2016; He et al., 2022a), and could be of independent interest.

The proofs of the theorems and corollaries of this section can be found in Appendix B.

## 4 Generalized Directed Hypergraph Neural Network (GeDi-HNN)

We embed the Generalized Directed Laplacian $\vec{L}_N$ in GeDi-HNN, the first HNN capable of handling hypergraphs with both undirected and directed hyperedges via a spectral-based convolution operator. For this purpose, we rely on the localized filter of Section 2 which led to equation 1. Letting $\mathcal{L} = \vec{L}_N$, our convolution operator is $\hat{Y}x = \theta_0 I + \theta_1 \vec{L}_N$.

The adoption of a localized filter with two parameters $\theta_0, \theta_1$ plays an important role towards the generality and the flexibility of GeDi-HNN, which we highlight next:

**Proposition 1.** *The convolution operator obtained from equation 1 by letting $\mathcal{L} = \vec{L}_N$ with parameters $\theta_0, \theta_1$ coincides with the one obtained by letting $\mathcal{L} = \vec{Q}_N$ with parameters $\theta_0' = \theta_0 + \theta_1$, $\theta_1' = -\theta_1$.*

The proposition implies that GeDi-HNN generalizes previously-proposed GNNs and HNNs irrespective of whether they are designed around a Signed or a Signless Laplacian matrix (provided that either $\vec{L}_N$ or $\vec{Q}_N$ generalize the matrix such networks employ). This is because, as the proposition shows, GeDi-HNN can implement the convolution of equation 1 built on top of either Laplacians by learning suitable values for $\theta_0, \theta_1$.

In our implementation, GeDi-HNN features the following extension of the convolution operator of equation 1. Let $X \in \mathbb{C}^{n \times c_0}$ be a $c_0$-dimensional graph signal (a graph signal with $c_0$ input channels), which we compactly represent as a matrix. We combine $\theta_0$ and $\theta_1$ with the *mixing* operator that is commonly applied to $X$ to linearly combine the $c_0$ channels of the graph signal. In doing so, we introduce two linear operators $\Theta_0, \Theta_1 \in \mathbb{C}^{c_0 \times c}$ which can either upscale (if $c > c_0$) or downscale (if $c < c_0$) the number of channels of $X$. A similar choice is made in other GNN/HNNs such as MagNet (Zhang et al., 2021b).

Letting $\phi$ be an activation function applied component-wise to its input matrix, the output $Z \in \mathbb{C}^{n \times c'}$ of any of GeDi-HNN's convolutional layers is:

$$Z(X) = \phi \left( IX\Theta_0 + \vec{L}_N X \Theta_1 \right). \tag{11}$$

As activation function, we adopt a complex extension of the *ReLU* function, in which, for a given $z \in \mathbb{C}$, $\phi(z) = z$ if $\Re(z) \geq 0$ and $\phi(z) = 0$ otherwise. A similar choice is followed in other GNNs/HNNs works such as (Fiorini et al., 2023; 2024). We project the complex-valued output of the last convolutional layer into the reals via an *unwind* operation by which $Z(X) \in \mathbb{C}^{n \times c}$ is transformed into $(\Re(Z(X))||\Im(Z(X))) \in \mathbb{R}^{n \times 2c}$, where $||$ is the concatenation operator.

To obtain the final result, we add $\ell$ linear layers to GeDi-HNN's architecture and a residual connection for every convolutional layer except for the first one. These connections have been proven to aid in training deeper models by allowing them to retain information from the input of the previous layers (He et al., 2016; Kipf and Welling, 2017). GeDi-HNN's architecture is depicted in Figure 1.

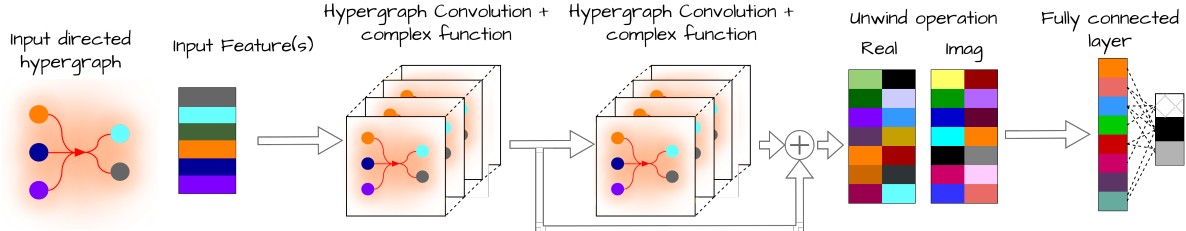

Figure 1: GeDi-HNN's architecture: after two complex convolutional layers and a residual connection, we unwind the real and imaginary parts of the feature matrix and apply a fully connected layer.

**Complexity of GeDi-HNN.** Let us assume w.l.o.g. (as it does not affect the asymptotic analysis of complexity) that the input and output data of each convolutional layer except for the last one have $c$ channels and that the output of the last convolutional layer and the number of channels of each linear layer are equal to $c'$. With $\ell$ convolutional layers and $S$ linear layers, GeDi-HNN's complexity is $O(\ell(n^2c + nc^2) + nc + (S - 1)(nc'^2) + nc'd + nd)$, where $d$ is the number of hidden node classes (the last channel of the last linear layer). Letting $\bar{c} := \max\{c, c', d\}$ be the largest number of channels throughout the network, we have a complexity of $O(\ell(n^2\bar{c}) + (\ell + S)(n\bar{c}^2))$. Such a quantity is quadratic w.r.t. the number of nodes $n$ and the largest number of channels $\bar{c}$, which is in line with previous GNNs and HNNs architectures. For more details see Appendix C.

## 5 Numerical Experiments

We now illustrate the results of an extensive set of experiments carried out to evaluate the performance of GeDi-HNN on directed hypergraphs. We compare our proposal against 11 state-of-the-art methods from the hypergraph-learning literature: HGNN (Feng et al., 2019), HCHA[4] (Bai et al., 2021), HCHA with the attention mechanism (Bai et al., 2021), HNHN (Dong et al., 2020), HyperGCN (Yadati et al., 2019), UniGCNII (Huang and Yang, 2021), HyperDN (Tudisco et al., 2021), AllDeepSets (Chien et al., 2021), AllSetTransformer (Chien et al., 2021), LEGCN Yang et al. (2022), ED-HNN (Wang et al., 2023a), and PhenomNN (Wang et al., 2023b). The hyperparameters of these baselines and of our proposed model are selected via grid search (see Appendix E).

The experiments are carried out on the node classification task of predicting the class associated with each node, which is the same task that was consistently used throughout the papers where the 11 baselines were proposed. The comparison is carried out on real-world datasets (Subsection 5.1) and on synthetic hypergraphs (Subsection 5.2).

Throughout the following tables, the best results are reported in **boldface** and the second-best are underlined. The datasets and code we used are publicly available on GitHub (see Appendix A).

---

[4]Among the many versions of HCHA in Dong et al. (2020), we use the one implemented in `https://github.com/Graph-COM/ED-HNN`, which coincides with HGNN$^+$ (Gao et al., 2022).

## 5.1 Node Classification Task on Real-World Datasets

We test GeDi-HNN on 10 real-world datasets from the literature: `Cora`, `Citeseer`, and `PubMed` (Zhang et al., 2022); `email-Enron` and `email-Eu` (Benson et al., 2018); `Texas`, `Wisconsin`, and `Cornell` (Pei et al., 2020); `WikiCS` (Mernyei and Cangea, 2020); and `Telegram` (Bovet and Grindrod, 2020). We test the 11 baselines (which, we recall, are not designed to handle directed hypergraphs) on an undirected version of the 10 datasets which is compiled following the procedure of Feng et al. (2019). Differently, we test GeDi-HNN, which is the only method designed to handle directed hypergraphs, on a directed version of these instances, which we compile following a slight modification to the previous procedure. For every node $p$ sharing a relationship with nodes $a, b, c, d$, we create the hyperedge $e$ with $H(e) = \{p\}$ and $T(e) = \{a, b, c, d\}$. Considering, e.g., a citation relationship in `CiteSeer` where paper $p$ cites papers $a, b, c, d$, in the undirected case we follow Feng et al. (2019) and create the hyperedge $\{a, b, c, d\}$ to semantically represent the paper $p$ whereas, in the directed case, we set $\{p\}$ as head and $\{a, b, c, d\}$ as tail. We adopt the split proposed by Zhang et al. (2021b) for `Telegram`, `Texas`, `Wisconsin`, and `Cornell` and the split of Chien et al. (2021) for the other ones. All the experiments are conducted using 10-fold cross-validation. More details on the datasets can be found in Appendix D.

Table 1: Mean accuracy and standard deviation obtained on the node classification task on the real-world datasets. The results of `Cora`, `Citeseer`, and `PubMed` are taken from Table 3 of (Wang et al., 2023b).

| Method | Cora | Citeseer | Pubmed | email-Eu | email-Enron |
|---|---|---|---|---|---|
| HGNN | $79.39 \pm 1.36$ | $72.45 \pm 1.16$ | $86.44 \pm 0.44$ | $39.80 \pm 2.77$ | $44.32 \pm 5.44$ |
| HCHA/HGNN$^+$ | $79.14 \pm 1.02$ | $72.42 \pm 1.42$ | $86.41 \pm 0.36$ | $41.01 \pm 3.55$ | $44.59 \pm 6.77$ |
| HCHA w/ Attention | $58.20 \pm 2.53$ | $68.44 \pm 1.27$ | $79.90 \pm 1.70$ | $28.75 \pm 2.82$ | $35.68 \pm 6.96$ |
| HNHN | $76.36 \pm 1.92$ | $72.64 \pm 1.57$ | $86.90 \pm 0.30$ | $29.92 \pm 1.88$ | $30.01 \pm 12.56$ |
| HyperGCN | $78.45 \pm 1.24$ | $71.28 \pm 0.82$ | $82.84 \pm 8.67$ | $30.81 \pm 2.80$ | $36.76 \pm 5.87$ |
| UniGCNII | $78.81 \pm 1.05$ | $73.05 \pm 2.21$ | $88.25 \pm 0.40$ | $40.81 \pm 2.76$ | $41.62 \pm 5.28$ |
| LEGCN | $74.74 \pm 1.25$ | $72.74 \pm 0.86$ | $88.12 \pm 0.74$ | $30.16 \pm 2.28$ | $35.41 \pm 5.76$ |
| HyperND | $79.20 \pm 1.14$ | $72.62 \pm 1.49$ | $86.68 \pm 0.43$ | $29.23 \pm 1.80$ | $35.41 \pm 5.62$ |
| AllDeepSets | $76.88 \pm 1.80$ | $70.83 \pm 1.63$ | $88.75 \pm 0.33$ | $29.92 \pm 1.88$ | $36.76 \pm 7.01$ |
| AllSetTransformer | $78.58 \pm 1.47$ | $73.08 \pm 1.20$ | $88.72 \pm 0.37$ | $\underline{41.58 \pm 5.13}$ | $\underline{45.41 \pm 8.43}$ |
| ED-HNN | $80.31 \pm 1.35$ | $73.70 \pm 1.38$ | $\underline{89.03 \pm 0.53}$ | $30.85 \pm 2.87$ | $42.97 \pm 7.37$ |
| PhenomNN | $82.29 \pm 1.42$ | $75.10 \pm 1.59$ | $88.07 \pm 0.48$ | $31.09 \pm 3.83$ | $37.03 \pm 7.21$ |
| **GeDi-HNN** | $\mathbf{84.04 \pm 1.15}$ | $\mathbf{75.68 \pm 1.04}$ | $\mathbf{89.80 \pm 0.51}$ | $\mathbf{49.27 \pm 3.17}$ | $\mathbf{52.43 \pm 5.28}$ |
| GeDi-HNN `w/o directions` | $78.85 \pm 1.75$ | $74.08 \pm 1.15$ | $88.67 \pm 0.58$ | $46.88 \pm 3.04$ | $48.38 \pm 6.55$ |

| Method | Telegram | Texas | Wisconsin | Cornell | WikiCS |
|---|---|---|---|---|---|
| HGNN | $\underline{59.42 \pm 6.04}$ | $71.08 \pm 7.32$ | $75.69 \pm 4.64$ | $70.81 \pm 4.73$ | $77.95 \pm 5.69$ |
| HCHA/HGNN$^+$ | $52.12 \pm 3.32$ | $71.35 \pm 6.77$ | $73.53 \pm 5.41$ | $70.81 \pm 5.06$ | $76.50 \pm 5.07$ |
| HCHA w/ attention | $57.69 \pm 2.86$ | $72.97 \pm 6.50$ | $70.59 \pm 6.40$ | $73.51 \pm 4.73$ | $11.67 \pm 4.79$ |
| HNHN | $50.77 \pm 8.27$ | $75.41 \pm 7.26$ | $81.18 \pm 3.72$ | $74.32 \pm 5.14$ | $26.47 \pm 18.1$ |
| HyperGCN | $55.77 \pm 3.95$ | $65.95 \pm 9.03$ | $70.98 \pm 5.05$ | $68.39 \pm 6.87$ | $75.80 \pm 6.16$ |
| UniGCNII | $55.58 \pm 5.01$ | $84.17 \pm 5.44$ | $86.47 \pm 5.02$ | $76.76 \pm 5.13$ | $83.24 \pm 1.07$ |
| LEGCN | $45.19 \pm 5.15$ | $\underline{79.19 \pm 4.78}$ | $84.51 \pm 5.35$ | $73.78 \pm 6.12$ | $\underline{78.73 \pm 1.19}$ |
| HyperND | $43.65 \pm 4.35$ | $81.62 \pm 6.60$ | $85.10 \pm 4.45$ | $74.87 \pm 4.60$ | $72.28 \pm 3.14$ |
| AllDeepSets | $38.46 \pm 6.08$ | $82.97 \pm 5.85$ | $84.51 \pm 5.43$ | $\underline{78.11 \pm 3.70}$ | $83.00 \pm 1.10$ |
| AllSetTransformer | $57.12 \pm 5.21$ | $80.27 \pm 5.56$ | $81.96 \pm 6.26$ | $76.47 \pm 5.41$ | $\mathbf{83.37 \pm 3.77}$ |
| ED-HNN | $54.42 \pm 6.01$ | $83.78 \pm 7.64$ | $86.27 \pm 2.45$ | $77.84 \pm 5.67$ | $82.12 \pm 1.57$ |
| PhenomNN | $54.61 \pm 4.72$ | $\mathbf{84.59 \pm 5.41}$ | $86.28 \pm 4.62$ | $76.49 \pm 5.56$ | $80.07 \pm 0.61$ |
| **GeDi-HNN** | $\mathbf{75.01 \pm 4.96}$ | $\mathbf{84.59 \pm 4.78}$ | $\mathbf{88.43 \pm 3.31}$ | $\mathbf{80.54 \pm 2.79}$ | $82.23 \pm 1.47$ |
| GeDi-HNN `w/o directions` | $64.80 \pm 6.60$ | $83.51 \pm 4.51$ | $86.66 \pm 4.96$ | $77.83 \pm 4.65$ | $82.52 \pm 1.19$ |

The accuracy obtained across the different methods and datasets is reported in Table 1. The results show that, across the whole dataset, GeDi-HNN achieves an average additive performance improvement over the best-performing competitor of approximately 4.20 percentage points. In terms of Relative Percentage Difference (RPD)[5], we have an average RPD improvement of 7.06%. The most significant improvement is observed on `Telegram`, where GeDi-HNN achieves an average RPD improvement of approximately 23.19% and an average additive improvement of 15.59 percentage points w.r.t. the best competitor from the literature (HGNN). Overall, GeDi-HNN ranks first on 9 out of 10 datasets and fourth on the 10th dataset, where it

---

[5]The RPD of two values $P_1, P_2$ is the percentage ratio of their difference to their average, i.e., $|P_1 - P_2| / \frac{P_1 + P_2}{2} \%$.

achieves an accuracy of 82.23, which is only 1.14 percentage points less than the best one recorded in the experiment.

Table 1 also presents the results of an ablation study aimed at demonstrating that a significant portion of the superior performance of GeDi-HNN is attributable to the Generalized Directed Laplacian $\vec{L}_N$ rather than to the network's architecture. In this study, we compare GeDi-HNN to GeDi-HNN `w/o directions`, a version which employs the undirected hypergraph Laplacian $\Delta$ proposed in Zhou et al. (2006) (which disregards hyperedge directions) instead of $\vec{L}_N$. As shown in Table 1, GeDi-HNN outperforms GeDi-HNN `w/o directions` with an RPD improvement of 4.90% (an additive difference of 3.35 percentage points) on average across 9 out of 10 datasets and achieves nearly identical performance on the 10th dataset (`WikiCS`), where the difference between the two versions is negligible (of, additively, only 0.29 percentage points). The largest improvement, observed on `Telegram`, is of a RPD of 14.61% (an additive difference of 10.21 percentage points).

These results underscore the importance of incorporating the directionality of the hyperedges and suggest that, thanks to the Generalized Directed Laplacian, GeDi-HNN effectively captures and utilizes this information.

## 5.2 Node Classification Task on Synthetic Datasets

To emphasize the importance of leveraging the hypergraph's directions in identifying the class of each node, we conduct a set of experiments on synthetic datasets specifically designed to exhibit a high degree of correlation between node classes and hyperedge directions.

Drawing inspiration from the methodology proposed in Zhang et al. (2021b), we rely on a collection of datasets which are generated as follows. First, the set $V$ of vertices is partitioned into $c$ equally-sized classes $C_1, \ldots, C_c$ with uniform probability. Subsequently, for each class $C$, $I_i$ intra-class undirected hyperedges are created, each with a cardinality uniformly sampled from $\{h_{\min}, \ldots, h_{\max}\}$, containing vertices of the same class also sampled with uniform probability. Similarly, for each pair of classes $C_i$ and $C_j$ with $i < j$, $I_o$ inter-class directed hyperedges are created. The head and tail sets are sampled from $C_i$ and $C_j$, respectively, with uniform probability, and both have a cardinality uniformly sampled from $\{h_{\min}, \ldots, h_{\max}\}$. Figure 2 portrays the directional flow the hyperedges induce among the classes of a synthetic dataset with 5 classes. Notice how the flow is directed from a class $C_i$ to a class $C_j$ only if $i < j$.

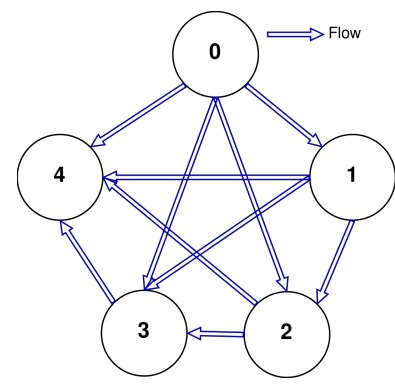

Figure 2: Schematic representation of the direction of the flow induced by the hyperedges in a synthetic dataset with 5 classes ($c = 5$).

Using this methodology, we generate three distinct datasets with parameters $n = 500$, $c = 5$, $h_{\min} = 3$, $h_{\max} = 10$, $I_i = 30$, and an increasing number of inter-class hyperedges $I_o = 10, 30, 50$. For these datasets, we implement a 50%/25%/25% split for training, validation, and testing, respectively. The experiments are conducted using 10-fold cross-validation.

The experimental results, summarized in Table 2, indicate a substantial performance difference between GeDi-HNN and the other 11 baselines, which increases the higher the value of $I_o$ is. On average, GeDi-HNN achieves an accuracy that surpasses the best competitor from the literature with an RPD improvement of 65.31% (an additive improvement of 35.47 percentage points). The additive difference compared to the second-best performer is of up to 39.19 percentage points.

We conclude with the ablation study where GeDi-HNN is compared to GeDi-HNN `w/o directions`. The results reveal a substantial accuracy difference of 40.48 percentage points (on average) between GeDi-HNN and GeDi-HNN `w/o directions` Notably; as the number of inter-class hyperedges ($I_o$) increases, the performance of GeDi-HNN improves, while that of GeDi-HNN `w/o directions` declines. This finding underscores the significant contribution of our proposed Generalized Directed Laplacian to the superior performance of GeDi-HNN.

Table 2: Mean accuracy and standard deviation obtained on the node classification tasks on the synthetic datasets.

| Method | $I_o = 10$ | $I_o = 30$ | $I_o = 50$ |
|---|---|---|---|
| HGNN | $30.02 \pm 5.99$ | $31.52 \pm 4.20$ | $32.40 \pm 3.36$ |
| HCHA/HGNN$^+$ | $33.60 \pm 4.76$ | $36.96 \pm 4.60$ | $39.04 \pm 2.66$ |
| HCHA w/ Attention | $17.28 \pm 2.42$ | $18.64 \pm 2.64$ | $20.44 \pm 2.24$ |
| HNHN | $19.28 \pm 4.16$ | $20.16 \pm 3.88$ | $19.28 \pm 2.86$ |
| HyperGCN | $21.04 \pm 3.99$ | $21.28 \pm 3.11$ | $17.84 \pm 3.33$ |
| UniGCNII | $20.80 \pm 3.94$ | $21.52 \pm 3.72$ | $20.40 \pm 4.67$ |
| LEGCN | $17.84 \pm 1.31$ | $19.76 \pm 5.27$ | $19.84 \pm 4.04$ |
| HyperND | $18.16 \pm 3.11$ | $18.40 \pm 3.85$ | $18.16 \pm 4.06$ |
| AllDeepSets | $18.32 \pm 4.12$ | $19.20 \pm 4.33$ | $18.72 \pm 4.40$ |
| AllSetTransformer | $19.44 \pm 4.42$ | $18.96 \pm 4.30$ | $22.72 \pm 5.06$ |
| ED-HNN | $19.12 \pm 3.32$ | $21.12 \pm 3.56$ | $18.96 \pm 3.33$ |
| PhenomNN | $20.88 \pm 4.24$ | $21.20 \pm 4.30$ | $20.32 \pm 4.62$ |
| **GeDi-HNN** | $\mathbf{65.92 \pm 3.98}$ | $\mathbf{71.84 \pm 3.31}$ | $\mathbf{78.24 \pm 5.64}$ |
| GeDi-HNN `w/o directions` | $36.72 \pm 5.84$ | $29.68 \pm 6.78$ | $28.16 \pm 10.65$ |

## 6 Conclusion

We introduced GeDi-HNN, the first spectral HNN capable of handling hypergraphs with both undirected and directed edges. GeDi-HNN is built upon a novel complex-valued Laplacian matrix, the *Generalized Directed Laplacian*, which is a Hermitian matrix that employs a complex-number representation of the hyperedge directions. This approach naturally generalizes several previously proposed Laplacians for both graphs and hypergraphs. Our proposal enables the seamless integration of directionality in HNNs, which is crucial for accurately modeling various real-world phenomena involving asymmetric high-order interactions. Our proposed GeDi-HNN model utilizes this new Laplacian matrix to perform spectral convolutions on hypergraphs featuring both undirected and directed hyperedges.

Extensive computational experiments on both real-world and synthetic datasets have demonstrated the superior performance of GeDi-HNN in 12 out of 13 experiments compared to a comprehensive representative group of state-of-the-art methods for the node classification task. These findings underscore the importance of incorporating directional information within GeDi-HNN's convolution operator. Specifically, GeDi-HNN consistently outperforms existing models across various datasets, achieving an average relative-percentage-difference improvement of 7% on real-world dataset (with a maximum improvement of 23.19%) and of 65.3% on synthetic datasets. The superiority of our method is particularly evident in the experiments on synthetic hypergraphs. These results highlight the potential of adopting GeDi-HNN to significantly enhance the modeling of complex, directed interactions within a hypergraph to the benefit of the hypergraph-learning task at hand.

### Broader Impact Statement

All the data we used are publicly available for research purposes and do not contain personally identifiable information or offensive content (see Appendix A for more details). The methods presented here have an impact on society comparable to other graph neural network algorithms.

### Acknowledgments

This work is supported by PNRR MUR Project Cod. PE0000013 "Future Artificial Intelligence Research (hereafter FAIR)" - CUP J53C22003010006.

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

## A  Code Repository and Licensing

The code written for this research work is available at `https://github.com/Stefa1994/GeDi-HNN` and freely distributed under the Apache 2.0 license.[6]

The `Texas`, `Wisconsin`, `Cornell`, `WikiCS`, and `Telegram` datasets were obtained from the PyTorch Geometric Signed Directed (He et al., 2022b) library (distributed under the MIT license). The `Cora`, `Citeseer`, and `PubMed` datasets are available at `https://linqs.org/datasets/`. The `email-Eu` and `email-Enron` datasets are available at `https://www.cs.cornell.edu/~arb/data/`.

The code for the baselines used in the experimental analysis is available at `https://github.com/Graph-COM/ED-HNN` and `https://github.com/yxzwang/PhenomNN` under the MIT license.[7]

## B  Properties of Our Proposed Laplacian $\vec{L}_N$

This section contains the proofs of the theorems, corollaries, propositions, and lemma reported in the main paper.

**Theorem 1.** *If $\mathcal{H}$ is an undirected hypergraph, $\vec{L}_N = \Delta$ and $\vec{Q}_N = Q_N$.*

*Proof.* Since $\mathcal{H} = (V, E)$ is an undirected hypergraph, $\vec{B}$ is binary and only takes values $0$ and $1$ (rather than being ternary and taking values $0, 1, -i$, which is the case in general). In particular, for each edge $e \in E$ we have $\vec{B}_{ue} = 1$ if either $u \in H(e)$ or $u \in T(e)$ and $\vec{B}_{ue} = 0$ otherwise. Consequently, the directed incidence matrix $\vec{B}$ is identical to the non-directed incidence matrix $B$, i.e., $\vec{B} = B$. Thus, by construction, $\vec{L}_N = \Delta$ and $\vec{Q}_N = Q_N$. $\qquad\square$

**Corollary 1.** *If $\mathcal{H}$ is an undirected 2-uniform hypergraph, $\vec{L}_N = \frac{1}{2}L_N$ and $\vec{Q}_N = \frac{1}{2}Q_N$.*

*Proof.* Since $\mathcal{H}$ is an undirected 2-uniform hypergraph, it follows that:

$$\begin{cases} \vec{B}W\vec{B}^* & = D_v + A \\ D_e^{-1} & = \frac{1}{2}I \end{cases}$$

Based on this, we can rewrite $\vec{Q}_N$ as follows:

$$\begin{aligned}
\vec{Q}_N &= D_v^{-\frac{1}{2}}\vec{B}WD_e^{-1}\vec{B}^*D_v^{-\frac{1}{2}} \\
&= D_v^{-\frac{1}{2}}\vec{B}\left(\frac{1}{2}W\right)\vec{B}^*D_v^{-\frac{1}{2}} \\
&= \frac{1}{2}\left(D_v^{-\frac{1}{2}}\left(D_v + A\right)D_v^{-\frac{1}{2}}\right) \\
&= \frac{1}{2}\left(I + D_v^{-\frac{1}{2}}AD_v^{-\frac{1}{2}}\right) \\
&= \frac{1}{2}\left(I + A_N\right) \\
&= \frac{1}{2}Q_N.
\end{aligned}$$

This proves the second part of the result. Since $\vec{Q}_N = \frac{1}{2}Q_N$ and, due to equation 5, $\frac{1}{2}L_N = I - \frac{1}{2}Q_N$, it follows that $\frac{1}{2}L_N = I - \vec{Q}_N = \vec{L}_N$. $\qquad\square$

**Theorem 2.** *If $\mathcal{H}$ is a directed 2-uniform hypergraph with no antiparallel edges, we have $\vec{L}_N = \frac{1}{2}L_N^\sigma$ with $A_s = A + A^\top$.*

---

[6]`https://www.apache.org/licenses/LICENSE-2.0`
[7]`https://choosealicense.com/licenses/mit/`

*Proof.* Since $\mathcal{H}$ is a directed 2-uniform hypergraph without antiparallel edges, it follows that:

$$\begin{cases} \vec{B}W\vec{B}^* = \bar{D}_s + H^\sigma \\ D_e^{-1} = \frac{1}{2}I. \end{cases}$$

Since $\mathcal{H}$ has no digons, the assumption $A_s = A + A^\top$ implies $\bar{D}_s = D_v$. Thus, we can rewrite $\vec{L}_N$ as follows:

$$\begin{aligned} \vec{L}_N &= I - D_v^{-\frac{1}{2}}\vec{B}W D_e^{-1}\vec{B}^* D_v^{-\frac{1}{2}} \\ &= I - D_v^{-\frac{1}{2}}\vec{B}\left(\frac{1}{2}W\right)\vec{B}^* D_v^{-\frac{1}{2}} \\ &= I - \frac{1}{2}\left(D_v^{-\frac{1}{2}}\left(\bar{D}_s + H^\sigma\right)D_v^{-\frac{1}{2}}\right) \\ &= I - \frac{1}{2}\left(I + D_v^{-\frac{1}{2}}H^\sigma D_v^{-\frac{1}{2}}\right) \\ &= \frac{1}{2}L_N^\sigma. \end{aligned}$$

$\square$

**Corollary 2.** *If $\mathcal{H}$ is a directed 2-uniform unweighted hypergraph with no antiparallel edges, we have $\vec{L}_N = \frac{1}{2}L_N^{(q)}$ with $q = \frac{1}{4}$ and $A_s = A + A^\top$.*

*Proof.* Since $\mathcal{H}$ is a directed 2-uniform unweighted hypergraph, $A \in \{0,1\}^{n \times n}$. Thus, as shown by Fiorini et al. (2023), with $q = \frac{1}{4}$ we have $L^\sigma = L^{(q)}$. Since Theorem 2 states that $\vec{L}_N = \frac{1}{2}L_N^\sigma$, it follows that

$$\vec{L}_N = \frac{1}{2}L_N^\sigma = \frac{1}{2}L_N^{(\frac{1}{4})}.$$

$\square$

**Theorem 3.** *$\vec{L}_N$ and $\vec{Q}_N$ are diagonalizable with real eigenvalues.*

*Proof.* This follows from the fact that the two matrices are, by construction, Hermitian. $\square$

**Theorem 4.** *$\vec{Q}_N$ is positive semidefinite.*

*Proof.*

$$\begin{aligned} x^*\vec{Q}_N x := & x^*\left(D_v^{-\frac{1}{2}}\vec{B}W D_e^{-1}\vec{B}^* D_v^{-\frac{1}{2}}\right)x \\ & \left(x^* D_v^{-\frac{1}{2}}\vec{B}W^{\frac{1}{2}} D_e^{-\frac{1}{2}}\right)\left(D_e^{-\frac{1}{2}}W^{\frac{1}{2}}\vec{B}^* D_v^{-\frac{1}{2}}x\right) \\ & \left(D_e^{-\frac{1}{2}}W^{\frac{1}{2}}\vec{B}^* D_v^{-\frac{1}{2}}x\right)^*\left(D_e^{-\frac{1}{2}}W^{\frac{1}{2}}\vec{B}^* D_v^{-\frac{1}{2}}x\right) \\ & \|\left(D_e^{-\frac{1}{2}}W^{\frac{1}{2}}\vec{B}^* D_v^{-\frac{1}{2}}x\right)^*\|_2^2 \geq 0. \end{aligned}$$

$\square$

**Theorem 5.** *Let $x = a + ib \in \mathbb{C}^n$, with $a, b \in \mathbb{R}^n$. The 2-Dirichlet energy function $\|x\|_{\vec{L}_N}^2 = x^*\vec{L}_N x$ of $x$ induced by $\vec{L}_N$ is the following quadratic form:*

$$\begin{aligned} \frac{1}{2}\sum_{e \in E}\frac{w(e)}{\delta(e)}\sum_{u,v \in E}&\left(\left(\left(\frac{a_u}{\sqrt{d_u}} - \frac{a_v}{\sqrt{d_v}}\right)^2 + \left(\frac{b_u}{\sqrt{d_u}} - \frac{b_v}{\sqrt{d_v}}\right)^2\right)\mathbf{1}_{u,v \in H(e) \vee u,v \in T(e)}\right. \\ &+ \left(\left(\frac{a_u}{\sqrt{d_u}} + \frac{b_v}{\sqrt{d_v}}\right)^2 + \left(\frac{a_v}{\sqrt{d_v}} - \frac{b_u}{\sqrt{d_u}}\right)^2\right)\mathbf{1}_{u \in H(e), v \in T(e)} \\ &\left.+ \left(\left(\frac{a_u}{\sqrt{d_u}} - \frac{b_v}{\sqrt{d_v}}\right)^2 + \left(\frac{a_v}{\sqrt{d_v}} + \frac{b_u}{\sqrt{d_u}}\right)^2\right)\mathbf{1}_{v \in H(e), u \in T(e)}\right), \end{aligned} \tag{12}$$

*where* **1** *is the indicator function.*

*Proof.*

$$x^* \vec{L}_N x = \sum_{u \in V} x_u^* x_u - \sum_{u,v \in V} \sum_{e \in E} \frac{w(e)}{\delta(e)} \frac{\bar{B}(u,e)\bar{B}(v,e)^*}{\sqrt{d(u)}\sqrt{d(v)}} x_u x_v^*$$

$$= \sum_{u \in V} x_u^* x_u - \sum_{e \in E} \sum_{u,v \in V} \frac{w(e)}{\delta(e)} \frac{\bar{B}(u,e)\bar{B}(v,e)^*}{\sqrt{d(u)}\sqrt{d(v)}} x_u x_v^*$$

$$= \sum_{u \in V} x_u^* x_u - \sum_{e \in E} \frac{w(e)}{\delta(e)} \sum_{u,v \in V: u \leq v} \left( \bar{B}(u,e)\bar{B}(v,e)^* \frac{x_u x_v^*}{\sqrt{d(u)}\sqrt{d(v)}} + \bar{B}(v,e)\bar{B}(u,e)^* \frac{x_v x_u^*}{\sqrt{d(v)}\sqrt{d(u)}} \right)$$

$$= \sum_{e \in E} \frac{w(e)}{\delta(e)} \sum_{u,v \in E: u \leq v} \left( \frac{x_u^* x_u}{d(u)} + \frac{x_v^* x_v}{d(v)} \right)$$

$$- \sum_{e \in E} \frac{w(e)}{\delta(e)} \sum_{u,v \in V: u \leq v} \left( \bar{B}(u,e)\bar{B}(v,e)^* \frac{x_u x_v^*}{\sqrt{d(u)}\sqrt{d(v)}} + \bar{B}(v,e)\bar{B}(u,e)^* \frac{x_v x_u^*}{\sqrt{d(v)}\sqrt{d(u)}} \right)$$

$$= \sum_{e \in E} \frac{w(e)}{\delta(e)} \sum_{u,v \in V: u \leq v} \left( \frac{x_u^* x_u}{d(u)} + \frac{x_v^* x_v}{d(v)} - \bar{B}(u,e)\bar{B}(v,e)^* \frac{x_u x_v^*}{\sqrt{d(u)}\sqrt{d(v)}} - \bar{B}(v,e)\bar{B}(u,e)^* \frac{x_v x_u^*}{\sqrt{d(v)}\sqrt{d(u)}} \right).$$

Let us analyze the three possible cases for the summand.

Case 1.a: $u \in H(e) \wedge v \in H(e) \Leftrightarrow \bar{B}(u,e) = 1, \bar{B}(v,e) = 1$. We have $\bar{B}(u,e)\bar{B}(v,e)^* = \bar{B}(v,e)\bar{B}(u,e)^* = 1$.

Case 1.b: $u \in T(e) \wedge v \in T(e) \Leftrightarrow \bar{B}(u,e) = -\mathrm{i}, \bar{B}(v,e) = -\mathrm{i}$. We have $\bar{B}(u,e)\bar{B}(v,e)^* = \bar{B}(v,e)\bar{B}(u,e)^* = (-\mathrm{i})(-\mathrm{i})^* = (-\mathrm{i})(\mathrm{i}) = 1$.

In both cases, we have:

$$\frac{x_u^* x_u}{d(u)} + \frac{x_v^* x_v}{d(v)} - \frac{x_u x_v^*}{\sqrt{d(u)}\sqrt{d(v)}} - \frac{x_v x_u^*}{\sqrt{d(v)}\sqrt{d(u)}} = \left( \frac{x_u}{\sqrt{d(u)}} - \frac{x_v}{\sqrt{d(v)}} \right)^* \left( \frac{x_u}{\sqrt{d(u)}} - \frac{x_v}{\sqrt{d(v)}} \right).$$

Letting $x_u = a_u + \mathrm{i}b_u$ and $x_v = a_v + \mathrm{i}b_v$, we have:

$$\left( \frac{a_u}{\sqrt{d_u}} - \frac{a_v}{\sqrt{d_v}} \right)^2 + \left( \frac{b_u}{\sqrt{d_u}} - \frac{b_v}{\sqrt{d_v}} \right)^2.$$

Case 2.a: $u \in H(e) \wedge v \in T(e) \Leftrightarrow \bar{B}(u,e) = 1, \bar{B}(v,e) = -\mathrm{i}$. We have $\bar{B}(u,e)\bar{B}(v,e)^* = (1)(-\mathrm{i})^* = \mathrm{i}$ and $\bar{B}(v,e)\bar{B}(u,e)^* = (-\mathrm{i})(1)^* = -\mathrm{i}$.

Thus:

$$\frac{x_u^* x_u}{d(u)} + \frac{x_v^* x_v}{d(v)} - \mathrm{i}\frac{x_u x_v^*}{\sqrt{d(u)}\sqrt{d(v)}} + \mathrm{i}\frac{x_v x_u^*}{\sqrt{d(v)}\sqrt{d(u)}}$$

Let $x_u = a_u + \mathrm{i}b_u$ and $x_v = a_v + \mathrm{i}b_v$, then we have:

$$\left( \frac{a_u}{\sqrt{d_u}} + \frac{b_v}{\sqrt{d_v}} \right)^2 + \left( \frac{a_v}{\sqrt{d_v}} - \frac{b_u}{\sqrt{d_u}} \right)^2.$$

Case 2.b: $u \in T(e) \wedge v \in H(e) \Leftrightarrow \bar{B}(u,e) = -\mathrm{i}, \bar{B}(v,e) = 1$. We have $\bar{B}(u,e)\bar{B}(v,e)^* = (-\mathrm{i})(1)^* = -\mathrm{i}$ and $\bar{B}(v,e)\bar{B}(u,e)^* = (1)(-\mathrm{i})^* = \mathrm{i}$. We have:

$$\frac{x_u^* x_u}{d(u)} + \frac{x_v^* x_v}{d(v)} + \mathrm{i}\frac{x_u x_v^*}{\sqrt{d(u)}\sqrt{d(v)}} - \mathrm{i}\frac{x_v x_u^*}{\sqrt{d(v)}\sqrt{d(u)}}$$

Let $x_u = a_u + ib_u$ and $x_v = a_v + ib_v$, then we have:

$$\left( \frac{a_u}{\sqrt{d_u}} - \frac{b_v}{\sqrt{d_v}} \right)^2 + \left( \frac{a_v}{\sqrt{d_v}} + \frac{b_u}{\sqrt{d_u}} \right)^2 .$$

The final equation reported in the statement of the theorem is obtained by combining the four cases we just analyzed. $\square$

**Corollary 3.** $\vec{L}_N$ *is positive semidefinite.*

*Proof.* Since $\bar{L}_N$ is Hermitian, it can be diagonalized as $U\Lambda U^*$ for some $U \in \mathbb{C}^{n \times n}$ and $\Lambda \in \mathbb{R}^{n \times n}$, where $\Lambda$ is diagonal and real. We have $x^*\bar{L}_N x = x^*U\Lambda U^*x = y^*\Lambda y$ with $y = U^*x$. Since $\Lambda$ is diagonal, we have $y^*\Lambda y = \sum_{u \in V} \lambda_u y_u^2$. Thanks to Theorem 5, the quadratic form $x^*\vec{L}_N x$ associated with $\vec{L}_N$ is a sum of squares and, hence, nonnegative. Combined with $x^*\bar{L}_N x = \sum_{u \in V} \lambda_u y_u^2$, we deduce $\lambda_u \geq 0$ for all $u \in V$. $\square$

**Corollary 4.** $\lambda_{\max}(\vec{L}_N) \leq 1$ *and* $\lambda_{\max}(\vec{Q}_N) \leq 1$.

*Proof.* $\lambda_{\max}(\vec{L}_N) \leq 1$ holds if and only if $\vec{L}_N - I \preceq 0$. Since $\vec{L}_N = I - \vec{Q}_N$ holds by definition, we need to prove $-\vec{Q}_N \preceq 0$, which holds true due to Theorem 4.

Similarly, $\lambda_{\max}(\vec{Q}_N) \leq 1$ holds if and only if $\vec{Q}_N - I \preceq 0$. Since $\vec{Q}_N = I - \vec{L}_N$ holds by definition, we need to prove $-\vec{L}_N \preceq 0$, which holds true due to Corollary 3. $\square$

**Proposition 1.** *The convolution operator obtained from equation 1 by letting $\mathcal{L} = \vec{L}_N$ with parameters $\theta_0, \theta_1$ coincides with the one obtained by letting $\mathcal{L} = \vec{Q}_N$ with parameters $\theta'_0 = \theta_0 + \theta_1$, $\theta'_1 = -\theta_1$.*

*Proof.* Consider the two operators $\theta_0 I + \theta_1 \vec{L}_N$ and $\theta'_0 I + \theta'_1 \vec{Q}_N$. Since $\vec{L}_N = I - \vec{Q}_N$, the first operator reads: $\theta_0 I + \theta_1(I - \vec{Q}_N)$. This is rewritten as $(\theta_0 + \theta_1)I - \theta_1 \vec{Q}_N$. By operating the choice $\theta'_0 = \theta_0 + \theta_1$ and $\theta_1 = -\theta'_1$, the second operator is obtained. $\square$

## C  Complexity of GeDi-HNN

The detailed calculations for the (inference) complexity of GeDi-HNN are as follows.

1. The Generalized Directed Laplacian $\vec{L}_N$ is constructed following equation 7 in time $O(n^2 m)$, where the factor $m$ is due to the need for computing the product between two rows of $\vec{B}$ to calculate each entry of $\vec{L}_N$. After $\vec{L}_N$ has been computed, the convolution matrix $\hat{Y} \in \mathbb{C}^{n \times n}$ is constructed in time $O(n^2)$. Note that such a construction is carried out entirely in pre-processing and is not required at inference time.

2. Each of the $\ell$ convolutional layers of GeDi-HNN requires $O(n^2c + nc^2 + nc) = O(n^2c + nc^2)$ elementary operations across 3 steps. Let $X^{l-1}$ be the input matrix to layer $l = 1, \ldots, \ell$. The operations that are carried out are the following ones.

   (a) $\vec{L}_N$ is multiplied by the node-feature matrix $X^{l-1} \in \mathbb{C}^{n \times c}$, obtaining $P^{l_1} \in \mathbb{C}^{n \times c}$ in time $O(n^2c)$ (we assume matrix multiplications takes cubic time);

   (b) The matrices $P^{l_0} = IX^{l-1} = X^{l-1}$ and $P^{l_1}$ are multiplied by the weight matrices $\Theta_0, \Theta_1 \in \mathbb{R}^{c \times c}$ (respectively), obtaining the intermediate matrices $P^{l_{01}}, P^{l_{11}} \in \mathbb{C}^{n \times c}$ in time $O(nc^2)$ .

   (c) The matrices $P^{l_{01}}$ and $P^{l_{11}}$ are additioned in time $O(nc)$ to obtain $P^{l_2}$.

   (d) The activation function $\phi$ is applied component-wise to $P^{l_2}$ in time $O(nc)$, resulting in the output matrix $X^l \in \mathbb{C}^{n \times c}$ of the $l$-th convolutional layer.

3. The unwind operator transforms $X^\ell$ (the output of the last convolutional layer $\ell$) into the matrix $U^0 \in \mathbb{R}^{n \times 2c}$ in linear time $O(nc)$.

4. Call $U^{s-1}$ the input matrix to each linear layer of index $s = 1, \ldots, S$. The application of the $s$-th linear layer to $U^{s-1} \in \mathbb{C}^{n \times c'}$ requires multiplying $U^{s-1}$ by a weight matrix $M_s \in \mathbb{C}^{c' \times c'}$ (where $c'$ is the number of channels from which and into which the feature vector of each node is projected). This is done in time $O(nc'^2)$.

5. In the last linear layer of index $S$, the input matrix $U^{S-1} \in \mathbb{R}^{n \times c'}$ is projected into the output matrix $O \in \mathbb{R}^{n \times d}$ in time $O(nc'd)$.

6. The application of the Softmax activation function takes linear time $O(nd)$.

We deduce an overall complexity of $O(\ell(n^2c + nc^2) + nc + (S-1)(nc'^2) + nc'd + nd)$ which, letting $\bar{c} = \max\{c, c', d\}$, coincides with $O(\ell(n^2\bar{c}) + (\ell + S)(n\bar{c}^2))$.

## D   Further Details on the Datasets

We test GeDi-HNN on ten real-world dataset. `Cora`, `Citeseer`, and `PubMed` (Zhang et al., 2022); `email-Eu`, and `email-Enron` (Benson et al., 2018); `Texas`, `Wisconsin`, and `Cornell` (Pei et al., 2020); `WikiCS` (Mernyei and Cangea, 2020); and `Telegram` (Bovet and Grindrod, 2020).

`Cora`, `Citeseer`, and `PubMed` are citation networks with node labels based on paper topics. In these citation networks, the nodes represent papers, their relationships denote citations of one paper by another, and the node features are the bag-of-words representation of papers.

`Email-Enron` and `email-Eu` are two email datasets—one from communications exchanged between Enron employees (Klimt and Yang, 2004) and the other from a European research institution (Paranjape et al., 2017). The nodes are email addresses and their relationships are of sender-receiver type. Since no node labeling is present in these two datasets, we define the node labels (node classes) using the Spinglass algorithm (Reichardt and Bornholdt, 2006).

`Texas`, `Wisconsin`, and `Cornell` are WebKB data sets extracted from the CMU World Wide Knowledge Base (`Web->KB`) project.[8] WebKB is a webpage data set collected from computer science departments of various universities by Carnegie Mellon University. In these networks, the nodes represent web pages, and the relationship are hyperlinks between them. The node features are the bag-of-words representation of the web pages. The web pages are manually classified into the five categories: student, project, course, staff, and faculty.

`WikiCS` is a directed network whose nodes correspond to Computer Science articles, and the relationships are on hyperlinks. This network has 10 classes representing different branches of the field.

`Telegram` models an influence network built on top of interactions among distinct users who propagate ideologies of a political nature.

The statistic of these ten real-world datasets and of the synthetic datasets we generate are summarized in Tables 3 and 4.

## E   Experiment Details

**Hardware.**   The experiments were conducted on 2 different machines:

1. An Intel(R) Xeon(R) Gold 6326 CPU @ 2.90GHz with 380 GB RAM, equipped with an NVIDIA Ampere A100 40GB.

2. A 12th Gen Intel(R) Core(TM) i9-12900KF CPU @ 3.20GHz CPU with 64 GB RAM, equipped with an NVIDIA RTX 4090 GPU.

---

[8]http://www.cs.cmu.edu/afs/cs.cmu.edu/project/theo-11/www/wwkb/

Table 3: Statistics of the real-world datasets

| Data set | # node | # hyperedges | # classes | average $|e|$ |
|---|---|---|---|---|
| Cora | 2708 | 1579 | 7 | 3.03 |
| Citeseer | 3312 | 1079 | 6 | 3.20 |
| Pubmed | 19717 | 7963 | 3 | 4.35 |
| email-Eu | 986 | 873 | 10 | 38.01 |
| email-Enron | 143 | 128 | 7 | 20.03 |
| Telegram | 245 | 185 | 4 | 48.04 |
| Texas | 183 | 40 | 5 | 4.45 |
| Wisconsin | 251 | 65 | 5 | 4.77 |
| Cornell | 183 | 41 | 5 | 3.88 |
| WikiCS | 11701 | 6827 | 10 | 42.08 |

Table 4: Statistics of the synthetic datasets

| Data set | # node | # hyperedges | # classes | average $|e|$ |
|---|---|---|---|---|
| $I_o = 10$ | 500 | 250 | 5 | 9.05 |
| $I_o = 30$ | 500 | 450 | 5 | 10.79 |
| $I_o = 50$ | 500 | 650 | 5 | 11.63 |

**Model Settings.** We trained every learning model considered in this paper for up to 500 epochs. We adopted a learning rate of $5 \cdot 10^{-3}$ and employed the optimization algorithm Adam with weight decays equal to $5 \cdot 10^{-4}$ (in order to avoid overfitting). For all the models that adopt the classification layer, we set it to 2.

We adopted a hyperparameter optimization procedure to identify the best set of parameters for each model. In particular, the hyperparameter values are:

- For AllDeepSets and ED-HNN, the number of basic block is chosen in $\{2, 4, 8\}$, the number of MLPs per block in $\{1, 2\}$, the dimension of the hidden MLP (i.e., the number of filters) in $\{64, 128, 256, 512\}$, and the classifier hidden dimension in $\{64, 128, 256\}$.

- For AllSetTransformer the number of basic block is chosen in $\{2, 4, 8\}$, the number of MLPs per block in $\{1, 2\}$, the dimension of the hidden MLP in $\{64, 128, 256, 512\}$, the classifier hidden dimension in $\{64, 128, 256\}$, and the number of heads in $\{1, 4, 8\}$.

- For UniGCNII, HGNN, HNHN, HCHA/HGNN$^+$, LEGCN, and HCHA with the attention mechanism, the number of basic blocks is chosen in $\{2, 4, 8\}$ and the hidden dimension of the MLP layer in $\{64, 128, 256, 512\}$.

- For HyperGCN, the number of basic blocks is chosen in $\{2, 4, 8\}$.

- For HyperND, the classifier hidden dimension is chosen in $\{64, 128, 256\}$.

- For PhenomNN, the number of basic blocks is chosen in $\{2, 4, 8\}$. We select four different settings:

  1. $\lambda_0 = 0.1$, $\lambda_1 = 0.1$ and prop step$= 8$,
  2. $\lambda_0 = 0$, $\lambda_1 = 50$ and prop step$= 16$,
  3. $\lambda_0 = 1$, $\lambda_1 = 1$ and prop step$= 16$,
  4. $\lambda_0 = 0$, $\lambda_1 = 20$ and prop step$= 16$.

- For GeDi-HNN and GeDi-HNN `w/o directionality`, the number of convolutional layers is chosen in $\{1, 2, 3\}$, the number of filters in $\{64, 128, 256, 512\}$, and the classifier hidden dimension in $\{64, 128, 256\}$. We tested GeDi-HNN both with the input feature matrix $X \in \mathbb{C}^{n \times c}$ where $\Re(X) = \Im(X) \neq 0$ and with $\Im(X) = 0$.

**Node Features.** For `Cora`, `Citeseer`, `PubMed`, `Texas`, `Wisconsin`, `Cornell`, `WikiCS`, and `Telegram`, we retain the datasets' original features. For `email-Eu`, `email-Enron`, and the synthetic datasets, the feature vectors are generated using the vertex degree of each node.

## F   From a Directed Hypergraph to the Generalized Directed Laplacian

To illustrate the representation of a directed hypergraph in our Generalized Directed Laplacian, consider a directed hypergraph $\mathcal{H} = (V, E)$ with $V = \{v_1, v_2, v_3, v_4, v_5\}$ and $E = \{e_1, e_2\}$. The incidence relationships are defined as follows: $v_1, v_2 \in H(e_1)$, $v_3 \in T(e_1)$, $v_4, v_5 \in H(e_2)$, and $v_1, v_2 \in T(e_2)$. The hyperedges have unit weights (i.e., $W = I$). The hyperedge cardinalities are $\delta_{e_1} = 3$ and $\delta_{e_2} = 4$.

For this hypergraph, we construct our Generalized Directed Laplacian using the following matrices: the incidence matrix $\vec{B}$, its conjugate transpose $\vec{B}^*$, the vertex degree matrix $D_v$, and the hyperedge degree matrix $D_e$.

$$\vec{B} = \begin{bmatrix} 1 & -i \\ 1 & -i \\ -i & 0 \\ 0 & 1 \\ 0 & 1 \end{bmatrix} \quad \vec{B}^* = \begin{bmatrix} 1 & 1 & i & 0 & 0 \\ i & i & 0 & 1 & 1 \end{bmatrix} \quad D_v = \begin{bmatrix} 2 & 0 & 0 & 0 & 0 \\ 0 & 2 & 0 & 0 & 0 \\ 0 & 0 & 1 & 0 & 0 \\ 0 & 0 & 0 & 1 & 0 \\ 0 & 0 & 0 & 0 & 1 \end{bmatrix} \quad D_e = \begin{bmatrix} 3 & 0 \\ 0 & 4 \end{bmatrix}.$$

Based on these matrices, we build $\vec{Q}_N$ as follows:

$$\vec{Q}_N = \begin{bmatrix} 0.29 & 0.29 & i0.24 & -i0.18 & -i0.18 \\ 0.29 & 0.29 & i0.24 & -i0.18 & -i0.18 \\ -i0.24 & -i0.24 & 0.33 & 0 & 0 \\ i0.18 & i0.18 & 0 & 0.25 & 0.25 \\ i0.18 & i0.18 & 0 & 0.25 & 0.25 \end{bmatrix}$$

and then our Generalized Directed Laplacian:

$$\vec{L}_N = \begin{bmatrix} 0.71 & -0.29 & -i0.24 & i0.18 & i0.18 \\ -0.29 & 0.71 & -i0.24 & i0.18 & i0.18 \\ i0.24 & i0.24 & 0.66 & 0 & 0 \\ -i0.18 & -i0.18 & 0 & 0.75 & -0.25 \\ -i0.18 & -i0.18 & 0 & -0.25 & 0.75 \end{bmatrix}$$

By inspecting $\vec{L}_N$, one can observe that it encodes the elements of the hypergraph in the following way:

1. The presence of nodes belonging to the same head or tail set, i.e., $v_1, v_2 \in H(e_1)$, $v_4, v_5 \in H(e_2)$, and $v_1, v_2 \in T(e_2)$, is encoded in the real part. Specifically, $(\vec{L}_N)_{v_1 v_2} = (\vec{L}_N)_{v_2 v_1} = -0.29$ and $(\vec{L}_N)_{v_4 v_5} = (\vec{L}_N)_{v_5 v_4} = -0.25$.

2. The directed hyperedges are encoded via the imaginary part. For example, considering nodes $v_1$ and $v_3$, we have $(\vec{L}_N)_{v_1 v_3} = -(\vec{L}_N)_{v_3 v_1} = -i0.24$.

3. The absence of a relationship between a pair of nodes is encoded by 0. Specifically, $(\vec{L}_N)_{v_3 v_4} = (\vec{L}_N)_{v_4 v_3} = 0$ and $(\vec{L}_N)_{v_3 v_5} = (\vec{L}_N)_{v_5 v_3} = 0$.

4. The "self-loop information" (a measure of how strongly the feature of a node depends on its current value within the convolution operator) is encoded by the diagonal of $\vec{L}_N$.

