# OpenReview forum: "Let There be Direction in Hypergraph Neural Networks"
_TMLR — Accepted by TMLR_

### Review · Reviewer_hfM7 · 2024-07-25

**Summary Of Contributions:**

The paper introduces a novel message passing technique for hypergraphs that allows to process both directed and undirected hyperedges simultaneously.
The method relies on a novel definition of the hypergraph Laplacian matrix that is proven to generalize previously introduced Laplacian matrices for undirected hypergraphs.
Specifically, the methodology is based on defining a complex-valued incidence matrix which allows the respective laplacian to capture the directed hyperedges in its imaginary part.
Furthermore, the authors establish a relationship between the undirected hypergraph Laplacian and verify that the matrix can indeed undergo spectral decomposition.
Finally, they propose a graph neural network based on a message passing on the proposed Laplacian matrix and concatenate the real and imaginary parts of the activations for the final output layer.
The method is evaluated by experiments on well-known datasets, indicating the superiority of the method compared to the literature of hypergraph NNs.

**Audience:**

Yes

**Broader Impact Concerns:**

There is no evident ethical concern about the paper.

**Claims And Evidence:**

Yes

**Requested Changes:**

As mentioned above, the paper could benefit from a comparison with simple GNNs on the regular datasets, to quantify the overall impact of adding hyperedges in this setting where they do not occur naturally.

Moreover, the authors could comment on any potential scalability issues that the method could encounter if it were tested with larger hypergraphs (e.g. DBLP and MAG datasets from https://www.cs.cornell.edu/~arb/data/).

**Strengths And Weaknesses:**

Strengths:

The idea is intuitive and sound.

The authors go in great lengths to position their Laplacian matrix with the literature e.g. the theorems linking it to the hypergraph Laplacian and the interpretation of Magnetic Laplacian with imaginary parts. This helps to motivate the method and clarify the contribution.

The outcome of the experiments clearly underlines the usefulness of considering the direction.

Weaknesses:

Although the definition in eq. 9 is clear and the examples in Appendix F are clear and useful, the inclusion of a scheme could facilitate conveying the message in the main paper.

To my understanding, part of the experiments on real-world datasets are based on regular undirected graphs that are enriched with hyperedges (e.g. citeseer), hence it would make sense if they are compared with regular GNNs as well (since the hyperedges are not integral to the dataset).

---

> ### Author Response · Authors · 2024-08-18
>
> Q: **[...] the paper could benefit from a comparison with simple GNNs on the regular datasets, to quantify the overall impact of adding hyperedges in this setting where they do not occur naturally. **
>
> A: Thank you for the comment. As per the reviewer’s request, we have conducted a comparison by applying four standard Graph Convolutional Networks (GCNs), namely ChebNet, GCN, SAGE, and GAT, to the regular datasets. The results are now presented in this table (where the best results are in **bold** and the second best are in *italics*):
>
> | Type      | Method     | Cornell        | Texas          | Wisconsin      | Cora           | CiteSeer       | Telegram       |
> |-----------|------------|----------------|----------------|----------------|----------------|----------------|----------------|
> | Spectral  | ChebNet    | 79.8%±5.0       | 79.2%±7.5       | _81.6%±6.3_       | 80.0%±1.8       | 66.7%±1.6       | 61.7%±4.2       |
> |           | GCN        | 59.0%±6.4       | 58.7%±3.8       | 55.9%±5.4       | _82.0%±1.1_       | 66.0%±1.5       | 60.8%±3.7       |
> | Spatial   | SAGE       | _80.0%±6.1_       | _84.3%±5.5_     | 83.1%±4.8     | 82.3%±1.2       | 66.0%±1.5       | 65.4%±5.2       |
> |           | GAT        | 57.6%±4.9       | 61.1%±5.0       | 54.1%±4.2       | 81.9%±1.0       | _67.3%±1.3_       | _72.6%±7.5_       |
> | Spectral  | GeDi-HNN   | **80.5%±2.8**   | **84.5%±4.8**   | **88.4%±3.3**   | **84.0%±1.2**   | **75.7%±1.1**   | **75.0%±5.1**   |
>
> | Type      | Method     | email-EU       | email-Enron    | Pubmed         | Wikics         |
> |-----------|------------|----------------|----------------|----------------|----------------|
> | Spectral  | ChebNet    | **55.6%±2.7**   | _67.2%±4.4_   | _89.1%±0.4_       | _83.0%±0.8_     |
> | Spatial   | SAGE       | 47.3%±5.1       | **67.6%±7.1**     | 88.5%±0.4       | **83.1%±2.1**   |
> |           | GAT        | _52.2%±4.4_     | 57.5%±8.1       | 86.1%±0.7       | 82.1%±1.1       |
> | Directed  | GeDi-HNN   | 49.3%±3.1       | 52.43%±5.3      | **89.8%±0.5**   | 82.2%±1.5       |
>
>
> As illustrated in the table, our model GeDi-HNN outperforms these standard GCNs on 7 out of the 10 datasets, while also achieving competitive results on the remaining ones. These findings underscore the positive impact of incorporating hyperedges into graphs in scenarios where hyperedges do not naturally occur. This further validates the strong performance of GeDi-HNN, which was already demonstrated in our initial comparisons against the 11 baselines included in the original submission.
>
> ---
>
> Q: **The authors could comment on any potential scalability issues that the method could encounter if it were tested with larger hypergraphs (e.g. DBLP and MAG datasets from https://www.cs.cornell.edu/ arb/data/).**
>
> A: Thanks for the comment. While the size of the Laplacian matrix is quadratic w.r.t. the number of nodes, scalability is rarely an issue if the graph is not too dense as one can always rely on a sparse matrix representations to encode the Laplacian and carry out algebraic operations with it---this is what we did in our code. Let us also remark that the inference complexity of our network is in line with previous GNNs/HNNs, as indicated in the paragraph "Complexity of GeDi-HNN at the end of Section 4. We will add a comment on the sparsity aspect to that paragraph to the revised version of the paper.

---

### Review · Reviewer_r6ZH · 2024-07-26

**Summary Of Contributions:**

The authors study the problem of learning on directed hypergraphs. They proposed a generalized directed hypergraph Laplacian operator and designed a hypergraph neural network corresponding to it. In theory, they show some desired properties of the proposed Laplacian operator and why it generalizes prior Laplacian operators. In practice, they conduct extensive experiments on both synthetic and real-world datasets to demonstrate the importance of both the direction of hyperedges and the new Laplacian operator for directed hypergraph learning.

**Audience:**

Yes

**Broader Impact Concerns:**

I do not foresee any broader impact concerns.

**Claims And Evidence:**

Yes

**Requested Changes:**

Please address my comments regarding the weaknesses if possible. Nevertheless, I am also fine with the manuscript in its current form.

**Strengths And Weaknesses:**

Strengths:

(+) In theory, the design of the new Laplacian operator seems reasonable. The discussion on its connection to prior Laplacian operators is extensive.

(+) In practice, the authors demonstrate superior learning performance for hypergraph node classification tasks.

(+) The authors also demonstrate the importance of the direction in hyperedges for learning tasks via experiments on both synthetic and real-world datasets.

Weaknesses:

(-) (minor) The paper reads a bit not concise, especially the discussion on the prior Laplacian operators.

(-) (minor) There are a few issues for the experiments, see comments below.

===

Overall I enjoy reading the manuscript. I feel the authors make a nice extension of the Laplacian operator to directed non-uniform hypergraphs. The experiments, especially those on the synthetic dataset, clearly demonstrate that the direction of hyperedges can be very helpful in the learning tasks.

There are some (minor) issues I found for the experiments, which may be resolved to strengthen the manuscript further. Firstly, the 10 real-world datasets are all undirected when they are initially introduced in the literature. The authors additionally construct a directed version of these datasets which technically speaking is not an apple-to-apple comparison. Nevertheless, the authors have made some effort to address this issue by testing their model with the original dataset as well. The comparison between GeDi-HNN versus its w/o directions counterpart at least demonstrates the importance of the direction in the hyperedges. I would suggest the authors further investigate (probably in future work) what datasets or real-world applications that inherently have these directions in the hyperedges. Then running it with GeDi-HNN versus ignoring the direction + standard HNN would better align with the claim made by the authors.

Second, I wonder why even applying GeDi-HNN w/o directions can already have better performance compared to standard HNN in general? Does that mean GeDi-HNN used in the experiments by default have more learnable weights? This makes me question if the number of learnable parameters of all methods is set to be similar so that it would be a fair comparison. I hope the authors can clarify this point.

Finally, there are some additional comments:

- The generic Laplacian matrix $\mathcal{L}$ is undefined for its first appearance. It should be at least explained briefly like $\mathcal{L}$ determined by the adjacency matrix.

- Why do we need $L_N = I-Q_N$ and $\Delta = I-Q_N$ when introducing what Zhou et al. (2006) did?

---

> ### Author Response · Authors · 2024-08-18
>
> Q: **The 10 real-world datasets are all undirected when they are initially introduced in the literature. The authors additionally construct a directed version of these datasets which technically speaking is not an apple-to-apple comparison. Nevertheless, the authors have made some effort to [...]**
>
> A: Thank you for your very insightful comment. We agree that exploring real-world applications with inherent directional hyperedges is important. To this end, we are currently investigating applications in molecular reactions, as highlighted in recent studies Mann and Venkatasubramanian (2023); Traversa et al. (2023). In these scenarios, directed hypergraphs naturally represent the transformation of a set of reagents (heads) into a set of products (tails), providing a meaningful context where hyperedge directionality plays a fundamental role. While these areas are indeed promising, they require a deep domain-specific understanding of the underlying chemical processes. Our preliminary results in this domain are encouraging, but we acknowledge that further work is needed before we can present any conclusive findings. In light of your suggestion, we will mentions these potential applications in the revised version of the paper.
>
> ---
>
> Q: **I wonder why even applying GeDi-HNN w/o directions can already have better performance compared to standard HNN in general? [...]**
>
> A: We would like to clarify the following points regarding the relation between the number of learnable weight and the performance of GeDi-HNN:
> - It is important to note that GeDi-HNN without directions does not consistently outperform standard HNNs. In fact, it performs worse than at least one of the competitors in 9 out of 13 instances.
> - We are confident in the fairness of our comparisons. We conducted extensive parameter tuning for each method, as detailed in Appendix E. The parameter tuning process adhered to the guidelines provided in the respective papers and code repositories. As a result, the number of learnable weights for each method was different but optimized through grid search to circumvent any unfair advantages a poor parameterization may lead to.
> - While it is true that GeDi-HNN has, on average, more learnable weights than many other HNNs in our experiments, our findings do not indicate that the number of parameters alone can lead to an improved performance. This is because GeDi-HNN without directions has the same number of weights as GeDi-HNN with directions but achieves a consistently worse performance (strictly worse in 9 cases out of 10). This indicates that the better performance of GeDi-HNN with directions is driven by its architectural design and its Laplacian rather than merely by the number of weights, highlighting the importance of how GeDi-HNN with directions’s architecture effectively leverages the hypergraph’s directions.
>
> We hope this clarification addresses your concerns and demonstrates that the comparisons in our experiments were conducted fairly and rigorously.
>
> ---
>
> Q: **The generic Laplacian matrix $\mathcal{L}$ is undefined for its first appearance. It should be at least explained briefly L like determined by the adjacency matrix.**
>
> A: Thank you for the suggestion. In the revised version of the paper, we will extend the sentence you mentioned as follows: "Let $\mathcal{L}$ be a generic Laplacian matrix of a given 2-uniform hypergraph $\mathcal{H}$, i.e., a matrix representation of the graph which encodes its connectivity and hyperedge weights—in the basic case where the hypergraph is not just 2-uniform but also undirected, $\mathcal{L}$ is defined as the difference between the degree matrix and the adjacency matrix of the graph".
>
> ---
>
> Q: **Why do we need $L_N = I - Q_N$ and $\Delta = I - Q_N$ when introducing what Zhou et al. (2006) did?**
>
> A: Thanks for the comment. What we tried to explain is that the definitions of $L_N$ in (4) and (5) coincide for 2-uniform undirected hypergraphs, whereas they do not for not 2-uniform hypergraphs, where the definition that is used in the literature is (5).
> We introduced $\Delta$ since this is the symbol that was used the paper (Zhou et al. (2006)) where definition (5) was first adopted for general (not necessarily 2-uniform) hypergraphs. We admit, however, that this notation choice does not clarity the point very well. We will drop this notation in the revised version of the manuscript.

---

> > ### Comment · Reviewer_r6ZH · 2024-08-19
> >
> > I thank the authors for their response. I am satisfied and I wish all the best to the authors.

---

### Review · Reviewer_GHzt · 2024-08-04

**Summary Of Contributions:**

In the submitted manuscript, the authors propose a spectral hypergraph neural network (GeDi-HNN). They do so by proposing and theoretically studying a new graph shift operator: the generalised directed Laplacian. GeDi-HNN is demonstrated to outperform 11 baseline models on 10 datasets and in a simulation study.

**Audience:**

Yes

**Broader Impact Concerns:**

I do not have any concerns about this work that would necessitate a broader impact statement.

**Claims And Evidence:**

Yes

**Requested Changes:**

1] In the last equation of Section 2 you have the following term $\text{sign}(|A-A^T|)$. To me it seems that this term should always be one since you are taking the sign of an absolute value. Could you please elaborate on the meaning and benefit of this term? If the notation $|\cdot|$ refers to the matrix determinant here, then it may be valuable to define the dimension of the different terms in this equation involving the element-wise product.

2] In the discussion of Corollary 3 you mention that the positive semidefiniteness of $L_N$ is sufficient for its eigenvalues to be interpreted as graph frequencies. To me, this does not seem to be a valid conclusion to draw. It seems to me that for the eigenvalues to be interpreted as graph frequencies one also needs to consider the associated eigenvectors. Shuman et al. [1] for example did this when they recorded the zero-crossings of the Laplacian eigenvectors in their Figure 3. If you want to keep this conclusion in your paper, then a similar study may be helpful to you. Since it seems to me that the eigenvalues of any positive semidefinite graph shift operator cannot be interpreted as graph frequencies in general.

3] In Equation (11), where you define your GeDi-HNN you appear to be accounting for self-loops in the graph twice. Once explicitly via the term $IX\Theta_0$ and a second time implicitly in the term $L_NX\Theta_1$ since $L_N=I-Q_N.$ May it not be simpler (and better performing) to use $Q_N$ instead of $L_N$ in your model definition to only account for self-loops once with a single trainable weight matrix? My worry is that two separate trainable weights on self-loops destabilise your training process. It would be great if you could run some ablation studies on this topic.

4] Could you please report the sensitivity of your results to different initialisations for $\theta_0$ and $\theta_1$? Since these are trainable parameters with significant impact on your graph representation, I could imagine you being relatively sensitive to an advantageous initialisation of these.

5] Some of the 11 baselines that you consider in Section 5.1 may trivially extend to handle directed graphs. However, you simply consider their performance on the undirected graphs. Could you confirm that taking directed graphs as input is impossible for the 11 baselines? And if not, may it not make for a fairer model comparison to also consider the performance of these models on the directed graphs?

6] Minor Changes:

- In the defnition of $\bar{D}_s$ you use the notation $e$, which I did not see previously defined. Could you please define this notation or point me to where you already defined it previously?

- In Theorem 2 you discuss antiparallel edges, which I did not see previously defined. Could you please add their definition?

- In Theorem 2 you define $A_s = A+ A^T$. In Section 2 you use the same notation: $A_s = 1/2(A+ A^T).$ It may be marginally nicer to be consistent with the notation $A_s.$

- In Section 5.2 you write that you partition the set of vertices into equally-sized classes "with uniform probability". Since the sets are equally-sized and no edges are drawn at this stage, the sampling step may be redundant. I recommend removing the formulation with "with uniform probability" here.

- In Appendix F, the top right two entries of $B$ should be $-i$ instead of $i$ if I understood correctly. I furthermore noticed that the font for the imaginary number $i$ is different here in comparison to the main paper, e.g., Section 2.


[1] Shuman, D., S. Narang, Pascal Frossard, Antonio Ortega, and P. Vanderghenyst. "The Emerging Field of Signal Processing on Graphs." IEEE Signal Proc. Magazine (2013).

**Strengths And Weaknesses:**

Strengths

1] The manuscript is well-written and your ideas are clearly described.

2] The synthetic dataset generator defined in Section 5.2 is rather nice in my opinion. I suppose this is what a 'Directed Hypergraph Stochastic Blockmodel' may look like and I quite like it.

3] Considering 11 baseline methods is extensive and provides strong evidence in favour of your model.


Weaknesses

1] The baselines were all run on undirected graphs and seemingly no effort was made to construct a directed baseline.

2] It seems to me that several aspects need to be further specified and resolved for publications (see my Requested Changes).

---

> ### Author Response · Authors · 2024-08-18
>
> Q: **In the last equation of Section 2 you have the following term $sign(|A - A^T|)$ [...]**
>
> A: In $\text{sign}(|A - A^T|)$, the signum function and the absolute value are applied componentwise. Thus, $\text{sign}(|A - A^T|)$ produces a matrix of binary values, with 0 if $A_{uv} = A_{uv}^T$ and 1 if $A_{uv} \neq A_{uv}^T$. In the definition of $L^\sigma$, $\text{sign}(|A - A^T|)$ is used to nullify the real part of $L^\sigma$ when $A_{uv} \neq A_{uv}^\top$ (in which case the graph features a directed edge) and to retain the real part when $A_{uv} = A_{vu}^\top$ (in which case the graph features an undirected edge or, equivalently, two antiprallel edges of the same weight). We will better clarity this aspect in the revised version of the manuscript.
>
> ---
>
> Q: **In the discussion of Corollary 3 you mention that the positive semidefiniteness of $L_N$ is sufficient for its eigenvalues to be interpreted as graph frequencies. [...]**
>
> A: We agree on this. The study you proposed would indeed be interesting, albeit it would require a non-trivial extension since, besides employing hyperedges rather than edges, our Laplacian admits complex-valued eigenvectors, which would require an alternative definition of the notion of zero-crossing than Shuman et al.'s. We will drop the bit on interpreting the eigenvalues as graph frequencies and amend the paper thusly: "following the methods used in Kipf and Welling (2017); Zhang et al. (2021b); Fiorini et al. (2023), we show that our Laplacian is positive semidefinite and thus can be adopted as a convolutional operator."
>
> ---
>
> Q: **In Equation (11), where you define your GeDi-HNN you appear to be accounting for self-loops in the graph twice. [...]**
>
> A: Notice that, by construction, the diagonal entries of $Q_N$ and $L_N$ are identical (this holds even in the simpler case where the hypergraph is 2-uniform (i.e., a graph) and undirected). Therefore, both $Q_N$ and $L_N$ already contains the information about the self-loops on their diagonal and, thus, and using $Q_N$ instead of $L_N$ would lead to a convolution operator with the same diagonal entries. To address the reviewer's concern about the potential destabilization of the training process due to the double-counting of self-loops, we conducted some experiments with $\Theta_0 = 0$ to eliminate the double self-loop.
>
> | Method      | Cora         | Citeseer     | Pubmed       |
> |--------------------------|--------------|--------------|--------------|
> | GeDi-HNN w/ $\Theta_0$  and $\Theta_1$     | **84.04±1.15**   | **75.68±1.04**   | **89.80±0.51**   |
> | GeDi-HNN w/ $\Theta_1$           | 82.47±1.20   | 70.81±1.74   | 84.08±0.43   |
>
> The results with $\Theta_0 = 0$ are inferior to those obtained with both $\Theta_0$ and $\Theta_1$, suggesting that adopting separate trainable weights ($\Theta_0$ and $\Theta_1$) on the self-loops do not destabilize the training process but, actually, improves the performance.
>
> ---
>
> Q: **Could you please report the sensitivity of your results to different initialization for $\theta_0$ and $\theta_1$ [...]**
>
> A: Let us clarify that the parameters $\theta_0$ and $\theta_1$ are not hyperparameters of GeDi-HNN. In line with most of the GCN literature originated in the seminal work of Kipf and Welling (2017), these two parameters are generalized to the two linear operators $\Theta_0$ and $\Theta_1$ of learnable weights of size $c \times c'$, where $c$ and $c'$ are the input and output number of features (or channels) before and after the convolution. In our experiments, we follow Zhang et al. (2021b); He et al. (2022b); Fiorini et al. (2023; 2024) and initialize these matrices using Xavier Glorot’s initialization method. We will mention this in the revised version of the paper.
>
> ---
>
> Q: **Some of the 11 baselines that you consider in Section 5.1 may trivially extend to handle directed graphs. [...]**
>
> A: We confirm that the 11 baseline methods considered in Section 5.1 are not designed to manage the directionality in directed hypergraphs. Handling directed hypergraphs poses unique challenges that distinguish them from both undirected graphs and directed graphs both in the spectral and in the spatial case. Regarding spectral approaches, which rely on the graph Laplacian for their convolution operations, none of the existing Laplacian definitions is, to our knowledge, applicable to directed hypergraphs, as such definitions fail to capture the directionality of the hyperedges (which, in our proposed Laplacian, is encoded in the complex-valued incidence matrix $\vec B$). To address this, we are the first (to the best of our knowledge) to propose a hypergraph Laplacian matrix specifically designed for directed hypergraphs. Spatial methods are admittedly more flexible and easier to adapt due to the fact that they are not grounded in a mathematical theory as solid as spectral methods but, in spite of this, also for this class we are not aware of any proposals suitably designed to handle directed hypergraphs.

---

> > ### Comment · Reviewer_GHzt · 2024-08-29
> >
> > I would like to thank the authors for their answers. Several of my concerns are resolved by their replies. However, there appear to be two cases in which my comments are incompletely addressed. I list these below.
> >
> > 1] Thanks for your further explanation. I may not have been sufficiently specific in my question. I am really wondering whether $\text{sign}(|A-A^T|) = |A-A^T|.$ To me this equality seems to hold and therefore, the $\text{sign}(\cdot)$ function should be redundant in your definition. Is this wrong?
> >
> > 3.1] In your answer you say that "by construction, the diagonal entries of $Q_N$ and $L_N$ are identical". This seems to be false to me, in Equation (7) of your paper you write $L_N = I - Q_N.$ Hence, (L_N)\_{ii} = 1 - (Q_N)\_{ii}. I therefore, feel that the self-loop information in the two operators should be different. Could you please clarify this issue?
> >
> > 3.2] I furthermore feel that your ablation studies conflate the presence/absence of skip-connections with the use of the two operators, i.e., $\Theta_0=0$ implies the absence of a skip-connection. You are studying the effect of having a single trainable weight matrix on both the self-loops and neighbourhood information when $\Theta_0=0$ and comparing this to a situation in which you have the trainable weight matrix $\Theta_1$ applied to both the self-loops and neighbourhood information, as well as matrix $\Theta_0$ applied to the self-loops once more. It seems to me that to truly study the impact of using $Q_N$ in your model definition, as I suggested, you would need to compare, the your current formulation $IX\Theta_0 + L_N X \Theta_1$ with $IX\Theta_0 + Q_N X \Theta_1.$

---

> ### Author Response · Authors · 2024-09-04
>
> **Q: [...] I am really wondering whether $sign(|A-A^\top|) = | A - A^\top|$.**
>
> A: Let us note that $sign(|A-A^\top|) = | A - A^\top|$ holds only when the identity matrix is binary, i.e.,$A \in \\{0, 1\\}^{n \times n}$. Otherwise, if $A$ has real values (which is the case for weighted graphs, for which the Laplacian matrix of Equation of $L^\sigma$ was proposed), the two expressions do not coincide.
>
> ---
>
> **Q: [...] I therefore, feel that the self-loop information in the two operators should be different. Could you please clarify this issue?**
>
> A: We confirm that the reviewer is right and that diagonals of the two matrices are indeed different -- what we meant to write was that the two diagonals are identical in the 2-uniform case (and not "even in" such a case) since, in it, both feature diagonals with identical elements all of value $\frac{1}{2}$. The self-loop information is indeed different, as one can see from the numerical example reported in Appendix F in the paper.
>
> ---
>
> **Q: I furthermore feel that your ablation studies conflate the presence/absence of skip-connections with the use of the two operators, i.e., $\Theta_0 = 0$ implies the absence of a skip-connection. [...] It seems to me that to truly study the impact of using $Q_N$ in your model definition, as I suggested, you would need to compare, your current formulation $IX\Theta_0 + L_NX\vec \Theta_1$ with $IX\Theta_0 + Q_NX\vec \Theta_1$.**
>
> A: As requested, we present below the results obtained from comparing the current formulation which employs the equation $IX\vec\Theta_0 + L_NX\vec\Theta_1$, with the suggested alternative formulation employing the equation $IX\vec\Theta_0 + Q_NX\vec\Theta_1$.
>
> The findings indicate that the performance obtained by using the original equation is essentially comparable to the one obtained with the alternative equation, with the results obtained with $IX\vec\Theta_0 + L_NX\vec\Theta_1$ being marginally better.
>
> | Method             | Cora               | Citeseer           | Pubmed            | email-EU          | email-Enron       |
> |--------------------|--------------------|--------------------|-------------------|-------------------|-------------------|
> | **GeDi-HNN w/ $L_N$** | **84.04±1.15**     | **75.68±1.04**     | **89.80±0.51**    | **49.27±3.17**    | **52.43±5.28**    |
> | GeDi-HNN w/ $Q_N$    | 78.52±1.55         | 72.05±1.16         | 89.11±0.53        | 49.23±2.60        | 52.16±6.50        |
>
> | Method             | Telegram           | Texas              | Wisconsin         | Cornell           | WikiCS            |
> |--------------------|--------------------|--------------------|-------------------|-------------------|-------------------|
> | **GeDi-HNN w/ $L_N$** | **75.01±4.96**     | **84.59±4.78**     | **88.43±3.31**    | **80.54±2.79**    | **82.23±1.47**    |
> | GeDi-HNN w/ $Q_N$    | 70.48±5.84         | 80.85±6.18         | 87.50±4.54        | 79.36±6.27        | 80.14±1.34        |
>
> Let us note that in the equations of our network the skip-connection shows up as shown in the following equations:
>
> $X^{1} = \phi\left(IX^l \Theta_0^0 + \vec{L}_N X^0 \Theta_1^0 \right)$
>
> $X^{2} = X^{1} \underbrace{+}_{\text{skip conn.}} \phi\left(IX^{1} \Theta_0^{1} + \vec{L}_N X^{1} \Theta_1^{1} \right)$
>
> where: $\Theta_0^0, \Theta_1^0$ are the $\Theta_0, \Theta_1$ matrices of the first layer; $\Theta_0^1, \Theta_1^1$ are the $\Theta_0, \Theta_1$ matrices of the second layer; $X_0$ is the input feature matrix; and $X^1,X^2$ are, respectively, the output feature matrices of convolutional layers 1 and 2.
>
> Let us notice that, with this architecture, even when setting \(\Theta_0 = 0\) the skip-connection remains unaffected since the equations become:
>
> $X^{1} = \phi\left(\vec{L}_N X^0 \Theta_1^0 \right)$
>
> $X^{2} = X^{1} \underbrace{+}_{\text{skip conn.}} \phi\left(\vec{L}_N X^{1} \Theta_1^{1} \right).$

---

> > ### Comment · Reviewer_GHzt · 2024-09-06
> >
> > I want to thank the authors for their careful and insightful responses. I am particularly thankful for the additional experimental results that were provided. My concerns have been completely resolved.

---

### Decision · Action_Editor_4Cip · 2024-09-13

**Recommendation:** Accept as is

**Comment:**

The paper was reviewed by three expert reviewers. All the reviewers acknowledged the technical contribution of the work. Some concerns were raised mainly about the fairness of the experiments and the baselines (e.g., whether they can be generalized to handle directed hypergraphs). The authors provided clarifications in their response, and all reviewers eventually recommended acceptance of the paper. I also agree with the reviewers and feel that the paper is ready for publication at TMLR.

**Audience:**

Since many real-world data can be represented as hypergraphs whose hyperedges have a notion of direction, the findings of this paper will be of interest to some individuals in TMLR's audience.

**Claims And Evidence:**

This paper proposes a graph neural network model which can handle hypergraphs that contain both directed and undirected hyperedges. To achieve that, the model employs a convolution operator which is built on top of a  complex-valued Hermitian Laplacian matrix. Both the proposed model and the use of the convolution operator seem reasonable. The paper's main claim is supported by the reported empirical results on synthetic datasets.